# A Semi-smooth, Self-shifting, and Singular Newton Method for Sparse Optimal Transport

## Abstract

Newton's method is an important second-order optimization algorithm that has been extensively studied. However, many challenging optimization problems break the classical assumptions of Newton's method. For example, the objective function may not be twice differentiable, and the optimal solution may be non-unique. In this article, we propose a general Newton-type algorithm named S5N, to solve problems that have possibly non-smooth gradients and non-isolated solutions, a setting highly motivated by the sparse optimal transport problem. Compared with existing Newton-type approaches, the proposed S5N algorithm has broad applicability, does not require hyperparameter tuning, and possesses rigorous global and local convergence guarantees. Extensive numerical experiments show that on sparse optimal transport problems, S5N gains superior performance on convergence speed and computational efficiency.

## 1 Introduction

Optimal transport (OT, Villani et al., 2009) is an important tool for modern machine learning, as it characterizes how one probability measure can be transformed to another, and it defines the Wasserstein distance as a metric between statistical distributions. OT also motivates many impactful machine learning models such as the Wasserstein generative adversarial networks (Arjovsky et al., 2017), differentiable sorting and nearest neighbors (Cuturi et al., 2019; Xie et al., 2020), and OT-based domain adaptation (Courty et al., 2017), among many others. One major challenge of applying OT to large-scale problems is its high computational cost, so various approximate OT algorithms have been proposed, among which one of the most popular approaches is the entropic-regularized OT based on the well-known Sinkhorn algorithm (Cuturi, 2013). However, the transport map obtained from the Sinkhorn algorithm is strictly positive, thus completely dense. In scenarios where the transport plan is of interest, a sparse result is generally preferred, as the unregularized optimal transport plan typically demonstrates a sparse pattern, and sparsity also enhances interpretability.

As an alternative approach, the quadratically regularized optimal transport (QROT, Blondel et al., 2018; Lorenz et al., 2021) outputs sparse transport plans. Proposition 1 of Blondel et al. (2018) shows that QROT can be reduced to an unconstrained optimization problem $\min_x f(x)$, where $f : \mathbb{R}^d \to \mathbb{R}$ is a continuously differentiable function, but $f$ is *not* twice continuously differentiable. Clearly, standard first-order methods such as the gradient descent algorithm can be used, but in general, second-order methods such as Newton's method have much faster convergence speed.

Newton's method is an important second-order optimization algorithm that has been extensively studied. Assuming $f$ is twice continuously differentiable, the classic Newton method generates a sequence of iterates $\{x^k\}$ using the updating formula $x^{k+1} = x^k - [H(x^k)]^{-1} g(x^k)$, where $g$ and $H$ are the first and second order derivatives of $f$, respectively.

However, for objective functions that are not twice differentiable, including the QROT dual objective function, the classic Newton method cannot be applied, as the Hessian matrix $H(x) = \nabla^2_{xx} f(x)$ is not well-defined. To this end, various generalized Newton methods, also known as the non-smooth Newton methods (Pang, 1990; Kummer, 1992; Qi & Sun, 1993; Qi, 1993), have been proposed, which take the form $x^{k+1} = x^k - V_k^{-1} g(x^k)$, where $V_k \in \mathbb{R}^{d \times d}$ is some substitute to the Hessian matrix $H(x^k)$ as in the smooth case; also see Qi & Sun (1999) for a survey of non-smooth Newton methods. For convenience, we loosely name $V_k$ as one generalized Hessian matrix of $f$ at $x^k$.

Although these non-smooth Newton methods have been successfully applied to many optimization problems, one critical limitation is that they require the matrix $V_k$ to be non-singular, which implies the uniqueness of the optimal point combined with other regularity conditions. Unfortunately, these conditions do not hold for QROT as well as many other challenging problems. In fact, Lorenz et al. (2021) proves that the generalized Hessian matrix $V_k$ for QROT has a non-trivial null space, and the optimal solution may not be unique or isolated.

Largely motivated by the QROT problem, in this article we propose a general Newton-type optimization method for objective functions with a semi-smooth gradient and a possibly singular generalized Hessian, where the definition of semi-smoothness is given in Section 2.1. The new method also has a mechanism to automatically adjust a shift parameter that needs to be manually set in some existing Newton-type algorithms, thus avoiding heavy hyperparameter tuning. Combining all these features, we name this algorithm as the semi-smooth, self-shifting, and singular Newton method (S5N).

We emphasize that although S5N is highly motivated by the sparse OT problem, the algorithm itself is fully general, and can be applied to many other problems. S5N borrows ideas from other second-order optimization techniques, including semi-smooth Newton methods (Qi, 1993; Qi & Sun, 1999), Levenberg–Marquardt methods (LM, Levenberg, 1944; Marquardt, 1963), and trust-region methods (Sorensen, 1982; Jorge & Stephen, 2006), but it combines the merits of these techniques in a non-trivial way, and also adds extra flexibility. We rigorously prove the global convergence of S5N, and also show that it has a fast quadratic local convergence rate under mild conditions. We compare S5N with various specialized algorithms for sparse OT on a large collection of test problems, and find that it consistently demonstrates a high performance both in convergence speed and actual run time.

Overall, our main contributions are summarized as follows:

1. We propose a novel Newton-type method, S5N, that is applicable to non-smooth gradients and singular (generalized) Hessian matrices, thus enriching the toolbox of second-order optimization algorithms for challenging problems.
2. We provide a rigorous convergence analysis for S5N, showing that it is globally convergent, and has a fast local quadratic convergence rate.
3. We test S5N on a large collection of sparse OT problems, and demonstrate its excellent computational efficiency and robustness across different data sets.
4. In addition to the methodology, we implement nine major sparse OT algorithms in efficient C++ code, and provide user-friendly Python and R packages.[1]

## 2 THE S5N METHOD

### 2.1 SEMI-SMOOTH FUNCTIONS

In this section, we revisit some fundamental results from non-smooth analysis that are necessary to develop the proposed algorithm. Throughout this article, $\|x\|$ is the Euclidean norm of a vector $x$, $\|A\|$ and $\|A\|_F$ stand for the operator norm and Frobenius norm of a matrix $A$, respectively, and $\langle A, B \rangle = \sum_i \sum_j a_{ij} b_{ij}$ represents the inner product between two matrices $A = (a_{ij})$ and $B = (b_{ij})$. We first introduce the notion of generalized differentials for functions that lack smooth derivatives, rooted in Rademacher's theorem.

**Theorem 1** (Rademacher). *Let $F: \mathbb{R}^n \to \mathbb{R}^m$ be a function that is locally Lipschitz continuous (definition in Appendix A). Then $F$ is differentiable almost everywhere.*

Denote by $D_F \subseteq \mathbb{R}^n$ the set of points at which $F$ is differentiable. Then the B(ouligand)-derivative and Clarke's generalized Jacobian are defined as follows.

**Definition 1** (Clarke, 1990). *Let $F : \mathbb{R}^n \to \mathbb{R}^m$ be locally Lipschitz continuous. The B-derivative of $F$ at point $x$ is defined as*

$$\partial_B F(x) = \left\{ \lim_{k \to \infty} \nabla F(x^k) \,\middle|\, x^k \in D_F, x^k \to x \right\}.$$

---

[1]The code to reproduce the experiments in this article is available at `https://1drv.ms/f/s!ArsORq8a24WmpE-MUCg5ph-uz-Pq`.

*Clarke's generalized Jacobian is then defined as the convex hull of the B-derivative, given by*

$$\partial F(x) = \mathrm{conv}(\partial_B F(x)).$$

B-derivative and Clarke's generalized Jacobian have the following properties.

**Theorem 2** (Clarke, 1990). *Let $F : \mathbb{R}^n \to \mathbb{R}^m$ be locally Lipschitz continuous. Then for $x \in \mathbb{R}^n$, the sets $\partial_B F(x)$ and $\partial F(x)$ are nonempty and compact, and the set-valued mappings $\partial_B F$ and $\partial F$ are locally bounded and upper semi-continuous (definition in Appendix A).*

Then we are ready to give the definition of semi-smooth functions.

**Definition 2** (Qi & Sun, 1993). *Let $F : \mathbb{R}^n \to \mathbb{R}^m$ be locally Lipschitz continuous. $F$ is said to be semi-smooth at point $x$ if for any $V \in \partial F(x + p)$,*

$$\|F(x + p) - F(x) - Vp\| = o(\|p\|), \quad as \ \|p\| \to 0.$$

*And $F$ is said to be strongly semi-smooth at $x$ if the condition above is replaced by*

$$\|F(x + p) - F(x) - Vp\| = O(\|p\|^2), \quad as \ \|p\| \to 0.$$

*If $F$ is (strongly) semi-smooth at any $x \in \mathbb{R}^n$, then it is said to be (strongly) semi-smooth on $\mathbb{R}^n$.*

## 2.2 THE ALGORITHM

Consider the unconstrained convex optimization problem,

$$\min_x \ f(x), \tag{1}$$

where $f : \mathbb{R}^d \to \mathbb{R}$ is convex and continuously differentiable. Let $g : \mathbb{R}^d \to \mathbb{R}^d$ be the first-order derivative of $f$, and $V \in \partial g(x)$ a generalized Jacobian (Hessian) matrix corresponding to $f$.

Similar to the classic Newton method, S5N generates a sequence of iterates $\{x^k\}$ to approach the optimal solution. In the $k$-th iteration, it updates the current iterate $x^k$ using information of $f_k = f(x^k)$, $g_k = g(x^k)$, and $V_k = V(x^k)$, where $V(x^k)$ is an arbitrary element from the generalized Jacobian $\partial g(x^k)$. Specifically, it computes a search direction $p^k$ of the form

$$p^k = -(V_k + \lambda_k I)^{-1} g_k, \tag{2}$$

where $\lambda_k > 0$ is a shift parameter whose expression will be given later. When $V_k$ is known to be positive semi-definite, which is the case when $f$ is convex, the matrix $V_k + \lambda_k I$ is always invertible. This is exactly the reason why S5N is applicable to singular $V_k$'s, whereas classic semi-smooth Newton methods are not. In addition, in each iteration, S5N computes a quantity $\rho_k$,

$$\rho_k = \frac{f(x^k) - f(x^k + \eta^k p^k)}{m(0) - m_k(\eta^k p^k)} = \frac{f_k - f(x^k + \eta^k p^k)}{f_k - m_k(\eta^k p^k)},$$

where $\eta^k > 0$ is a step size parameter, and the $m_k(\cdot)$ function is a local quadratic approximation to the objective function:

$$m_k(p) = f_k + g_k^T p + \frac{1}{2} p^T V_k p.$$

We postpone the discussion of $\eta^k$ parameters to Section 2.3, and for now the only requirement is that $\{\eta^k\}$ needs to be bounded from both below and above, *i.e.*, $\eta^k \in [\tilde{m}, \tilde{M}]$ for some $0 < \tilde{m} < \tilde{M} < \infty$.

Intuitively, $\rho_k$ measures the ratio between the actual and predicted reduction of function values. The quantity $\rho_k$ is commonly seen in trust-region methods, but there $\rho_k$ is typically defined by forcing $\eta^k \equiv 1$. In S5N, $\rho_k$ has two important functions: it determines how the shift parameter $\lambda_k$ is updated, and it decides whether to accept $x^k + \eta^k p^k$ as the next iterate. The full algorithm of S5N is given in Algorithm 1.

The design of Algorithm 1 is motivated by some widely-used second-order optimization methods. For example, the use of the generalized Hessian matrix $V_k$ comes from semi-smooth Newton methods, and the ratio $\rho_k$ is derived from trust-region methods. Moreover, the form of the shift parameter $\lambda_k = \mu_k \|g_k\|^\delta$ has been used by some LM-type algorithms such as Qi et al. (2016). Overall, S5N has the following advantages over existing Newton-type methods.

---

**Algorithm 1** A Semi-smooth, self-shifting, and singular Newton method (S5N)

1: **Input:** Initial point $x^0$, parameters $\{\mu_0, \tilde{m}, \tilde{M}, \kappa\} > 0$, $\rho_0 \in (0, \frac{1}{2})$, $\delta \in [1, 2]$, $\varepsilon_{tol} > 0$
2: **Default values:** $\mu_0 = 1$, $\tilde{m} = 0.01$, $\tilde{M} = 1$, $\kappa = 0.001$, $\rho_0 = \frac{1}{4}$, $\delta = 1$
3: **for** $k = 0, 1, 2, \dots$ **do**
4:     **if** $\|g_k\| < \varepsilon_{tol}$ **then return** $x^k$
5:     Compute $V_k \in \partial g(x^k)$, $p^k = -(V_k + \mu_k \|g_k\|^\delta I)^{-1} g_k$
6:     Select any $\eta^k \in [\tilde{m}, \tilde{M}]$
7:     Compute $\rho_k = \dfrac{f(x_k) - f(x_k + \eta^k p^k)}{m_k(0) - m_k(\eta^k p^k)}$
8:     Update $\mu_{k+1} = \begin{cases} 4\mu_k, & \text{if } \rho_k < \rho_0 \\ \max\{\mu_k/2, \kappa\}, & \text{if } \rho_k \geq 1 - \rho_0 \\ \mu_k, & \text{otherwise} \end{cases}$
9:     **if** $\rho_k > 0$ **then**
10:         $x^{k+1} = x^k + \eta^k p^k$
11:     **else**
12:         $x^{k+1} = x^k$

---

1. **Broad applicability**. S5N can handle both non-smooth gradients *and* singular (generalized) Hessian matrices, whereas many existing algorithms can only deal with one of the two issues.

2. **Solid convergence guarantees**. S5N has a rigorous global convergence guarantee without relying on line search procedures. It also has a fast quadratic local convergence rate.

3. **Tuning-free yet flexible**. Key algorithmic parameters (*e.g.*, $\lambda_k$) in S5N are automatically adjusted, but there is still flexibility in designing the step size parameter $\eta^k$ for better performance. The motivation and benefit of introducing $\eta^k$ is discussed in Section 2.3.

A summary of some existing Newton-type methods for non-smooth or singular problems is given in Table 1. We emphasize that combining the merits of existing algorithms and forming the S5N algorithm is non-trivial, since the convergence properties of S5N cannot be directly obtained by gathering existing results. For example, methods based on a non-singular Hessian rely on the uniqueness of the solution, whereas S5N has a completely different analysis.

Table 1: A summary of Newton-type methods for non-smooth or singular problems. Column (a): whether the method supports non-smooth gradients. Columns (b)(c): whether the algorithm itself and the theoretical analysis can proceed with singular Hessian matrices. Column (d): whether the method already has a global convergence guarantee, or relies on line search. Column (e): local convergence rate. Question marks mean that the article does not provide the corresponding analysis.

| | (a) Non-smooth gradient | (b) Singular Hessian (algorithm) | (c) Singular Hessian (theory) | (d) Global convergence | (e) Local convergence |
|---|---|---|---|---|---|
| Pang (1990); Qi (1993) | ✓ | ✗ | ✗ | Line search | Quadratic |
| Chen & Qi (1994) | ✓ | ✓ | ✗ | ? | Superlinear |
| Li et al. (2004) | ✗ | ✓ | ✓ | Line search | Quadratic |
| Zhou & Chen (2013) | ✗ | ✓ | ✓ | ✓ | Cubic |
| Zhou & Toh (2005) | ✓ | ✓ | ✓ | Line search | Superlinear |
| Xiao et al. (2018) | ✓ | ✓ | ✗ | ✓ | Quadratic |
| Lorenz et al. (2021) | ✓ | ✓ | ? | Line search | ? |
| This paper (S5N) | ✓ | ✓ | ✓ | ✓ | Quadratic |

## 2.3 Step Size Selection

The step size parameter $\eta^k$ in Algorithm 1 is a unique design for our S5N algorithm. On one hand, it is more flexible than classical trust-region methods, since in those algorithms the search direction $p^k$ is directly added to the current iterate $x^k$, *i.e.*, $x^{k+1} = x^k + p^k$. While trust-region methods offer an implicit step size adjustment mechanism via the $\lambda_k$ parameter, consistently using a unit step size often leads to repetitive sub-problem solving, which involves the computation of gradients, generalized Hessian, and linear systems, culminating in high computational costs.

On the other hand, unlike line-search-based methods, in S5N we do not require $\eta^k$ to satisfy any sufficient decrease or curvature conditions. In fact, the only requirement is that $\{\eta^k\}$ needs to be bounded from both below and above. As for line search approaches, their computational costs until termination are typically unpredictable, and to satisfy specific search conditions, infinite loops might occur if the algorithm is not carefully engineered. Moreover, line search methods typically require both objective function and gradient evaluations, which further increases the computational burden.

Based on these considerations, we advocate a step size selection method in Algorithm 2 to heuristically choose $\eta^k$ in each iteration. This method only sets the step size within a bounded range and computes objective function values for candidate step sizes. It is fully acceptable that in some iterations, the selected step size leads to an increase in objective function value, since the boundedness of $\eta^k$ still guarantees the global convergence, as we will show in Theorem 3. On the contrary, line search methods typically require $\eta^k$ to satisfy sufficient decrease or curvature conditions to guarantee global convergence. Due to this distinction, we position our method a step size *selection* scheme rather than a line *search* approach.

---

**Algorithm 2** Step size selection

---

1: **Input:** Candidate step sizes $1 = \eta_0 > \eta_1 > \cdots \eta_N > 0$, current $x^k$, $p^k$, objective function $f(\cdot)$
2: Initialize $\eta_{\text{best}} = 1$, $f_{\text{best}} = +\infty$
3: **for** $i = 0, 1, 2 \ldots, N$ **do**
4:     Compute $x_{\text{trial}} = x^k + \eta_i p^k$, $f_{\text{trial}} = f(x_{\text{trial}})$
5:     **if** $f_{\text{trial}} < f_{\text{best}}$ **then**
6:         $f_{\text{best}} = f_{\text{trial}}$
7:         $\eta_{\text{best}} = \eta_i$
8:     **if** $f_{\text{best}} < f(x^k)$ **then return** $\eta_{\text{best}}, f_{\text{best}}$
9: **return** $\eta_{\text{best}}, f_{\text{best}}$

---

## 2.4 Convergence Analysis

In this section, we present the main theorems for the convergence properties of S5N. While the assumptions and the presentation of the conclusions are similar to those of existing semi-smooth Newton methods (Zhou & Toh, 2005; Xiao et al., 2018), we emphasize that the proofs are specific to S5N under completely new settings, which are not trivial consequences of existing analyses. For global convergence, we make the following two assumptions.

**Assumption 1.** *The level set $L(x^0) = \{x \in \mathbb{R}^d : f(x) \le f(x^0)\}$ is bounded, and $f(x)$ is convex and continuously differentiable over $L(x^0)$. For all $x \in L(x^0)$, $f(x) > -\infty$.*

**Assumption 2.** *Let $L(x^0)$ be defined as in Assumption 1. For all $x \in L(x^0)$, $g(x)$ and any $V \in \partial g(x)$ satisfy*

$$\beta_1 = \max\left\{\|g(x)\| : x \in L(x^0)\right\} < \infty,$$
$$\beta_2 = \max\left\{\|V\| : V \in \partial g(x), x \in L(x^0)\right\} < \infty.$$

Then Theorem 3 shows that a subsequence of the iterates converges to an optimal solution.

**Theorem 3.** *Suppose that Assumptions 1 and 2 are satisfied, and let $\{x^k\}$ be generated by Algorithm 1. Then either Algorithm 1 terminates in finite iterations, or $g_k$ satisfies*

$$\liminf_{k \to \infty} \|g_k\| = 0.$$

Note that in our setting, the objective function $f(x)$ may have more than one optimal solution, and the solutions are not isolated due to the possible singularity of the generalized Hessian matrix. Therefore, it is hard to judge the behavior of the iterates $\{x^k\}$ purely from the global convergence result. However, under some extra conditions, we show that the iterates $\{x^k\}$ of the S5N algorithm will be attracted by some optimal point $x^*$ in the following sense: once $x^k$ enters a neighborhood of $x^*$, all subsequent iterates will stay in it, and $x^k$ then approaches the solution set $\mathcal{X}$ of problem (1) with a quadratic rate.

Let $\text{dist}(x, \mathcal{X}) = \inf_{y \in \mathcal{X}} \|y - x\|$ denote the distance between a point $x$ and the solution set $\mathcal{X}$, and $\bar{x}^k$ the projection of $x^k$ onto $\mathcal{X}$, *i.e.*,

$$\bar{x}^k = \arg \min_{\bar{x} \in \mathcal{X}} \|x^k - \bar{x}\|, \quad \|x^k - \bar{x}^k\| = \text{dist}(x^k, \mathcal{X}).$$

The notation $N(x, r) = \{y : \|y - x\| \leq r\}$ stands for the neighborhood of point $x$ with radius $r$. We then make the following additional assumptions.

**Assumption 3.** *Let $L(x^0)$ be defined as in Assumption 1, and $x^*$ an optimal point of problem (1).*

1. *$g$ is Lipschitz continuous and strongly semi-smooth over $L(x^0)$.*

2. *There exist constants $c_1, r_1 > 0$ such that $N(x^*, r_1) \subset L(x^0)$ and*

$$c_1 \cdot \text{dist}(x, \mathcal{X}) \leq \|g(x)\|, \quad \forall x \in N(x^*, r_1).$$

3. *There exists a constant $L > 0$ such that*

$$\|g(y) - g(x) - V(y - x)\| \leq L\|y - x\|^2, \quad \forall x, y \in N(x^*, r_1), V \in \partial g(x).$$

Then we have the local quadratic convergence guarantee as stated in Theorem 4.

**Theorem 4.** *Let Assumptions 1 to 3 be satisfied, and $\{x^k\}$ be generated by Algorithm 1. Then there exist constants $r, \tilde{c} > 0$ such that if for some integer $K$, $x^K \in N(x^*, r)$, then*

$$x^k \in N(x^*, r_1/2), \quad \forall k \geq K,$$

*and*

$$\text{dist}(x^k + \eta^k p^k, \mathcal{X}) \leq \tilde{c} \cdot \text{dist}(x^k, \mathcal{X})^2,$$

*which means that $\text{dist}(x^k, \mathcal{X}) \to 0$ quadratically.*

## 3 THE SPARSE OPTIMAL TRANSPORT PROBLEM

In this section, we briefly introduce the sparse OT problem, and show how it can be solved by the S5N algorithm. More background information for sparse OT and the QROT problem can be found in Blondel et al. (2018) and Lorenz et al. (2021).

First, define the probability simplex as $\Delta_n = \{a \in \mathbb{R}^n_+ : \sum_i a_i = 1\}$. Consider two discrete distributions $\nu \in \Delta_n$ and $\mu \in \Delta_m$, and let $U(\nu, \mu) = \{\Pi \in \mathbb{R}^{n \times m}_+ : \Pi \mathbf{1}_m = \nu, \Pi^T \mathbf{1}_n = \mu\}$ represent the space of couplings that form a joint distribution with $\nu$ and $\mu$ being the marginals. Then the QROT problem can be formulated as solving the optimization problem

$$W(\Pi; C, \nu, \mu) = \min_{\Pi \in U(\nu, \mu)} \langle C, \Pi \rangle + \frac{\gamma}{2} \|\Pi\|^2_F, \tag{3}$$

where $C \in \mathbb{R}^{n \times m}_+$ is a given cost matrix. Problem (3) has a unique global optimum $\Pi^*$, which is typically a sparse matrix. It can be viewed as a sparse approximation to the unregularized optimal transport plan (*i.e.*, the case with $\gamma = 0$).

Given two vectors $\alpha \in \mathbb{R}^n$ and $\beta \in \mathbb{R}^m$, define $\alpha \oplus \beta$ to be a matrix such that $(\alpha \oplus \beta)_{ij} = \alpha_i + \beta_j$. For a matrix $A = (a_{ij})$, $A_+$ stands for a matrix with elements $\max\{a_{ij}, 0\}$. Then Lorenz et al. (2021) shows that the dual problem of (3) is

$$\min_{x \in \mathbb{R}^{n+m}} f(x), \quad f(x) \equiv f(\alpha, \beta) = \frac{1}{2} \left\|(\alpha \oplus \beta - C)_+\right\|^2_F - \gamma \langle \nu, \alpha \rangle - \gamma \langle \mu, \beta \rangle, \tag{4}$$

where $x = (\alpha, \beta)$. Let $(\alpha^*, \beta^*)$ be any optimal solution to (4), and then the sparse transport plan $\Pi^*$ can be recovered as $\Pi^* = \gamma^{-1}(\alpha^* \oplus \beta^* - C)_+$.

Clearly, the function $f$ is continuously differentiable with respect to $x$, and we have the following expressions for its gradient and generalized Hessian matrix.

**Proposition 1.** *The gradient $g(\alpha, \beta)$ of $f(\alpha, \beta)$ is given by*

$$g(\alpha, \beta) = \begin{pmatrix} (\alpha \oplus \beta - C)_+ \mathbf{1}_m - \gamma\nu \\ \mathbf{1}_n^T(\alpha \oplus \beta - C)_+ - \gamma\mu \end{pmatrix},$$

*and $g$ is strongly semi-smooth. One generalized Hessian matrix of $f$ is*

$$V = \begin{pmatrix} \mathbf{diag}(\sigma \mathbf{1}_m) & \sigma \\ \sigma^T & \mathbf{diag}(\sigma^T \mathbf{1}_n) \end{pmatrix} \in \partial g(\alpha, \beta),$$

*where $\sigma_{ij} = 1$ if $\alpha_i + \beta_j - c_{ij} \geq 0$, and $\sigma_{ij} = 0$ otherwise. Moreover, $V$ is symmetric positive semi-definite.*

Given the gradient and the generalized Hessian matrix, we are then ready to use S5N to solve (4). In Algorithm 1, one key step is obtaining the Newton direction $p^k = -(V_k + \lambda_k I)^{-1} g_k$. Since $\sigma$ is in general a sparse matrix, the conjugate gradient method can be used to solve this linear system. We provide the computational details in Appendix C.

## 4 RELATED WORK

**Newton-type Methods** Besides the articles listed in Table 1, there are a few methods that handle the non-differentiability and singularity issues from a different angle. Chen et al. (1997) overcomes the singularity of $V_k$ by using its outer inverse, which, however, is typically obtained from costly matrix decomposition, such as QR decomposition and singular value decomposition. Facchinei et al. (2014) proposes the LP-Newton method, which applies to a class of non-smooth and singular problems with a local quadratic convergence guarantee. However, in each iteration, the algorithm needs to solve a large-scale linear programming problem, which can be very time-consuming.

**LM-type Methods** The LM algorithm was originally developed for nonlinear least squares problems, and later was also widely used for solving nonlinear equations of the form $F(x) = 0$, where $F : \mathbb{R}^n \to \mathbb{R}^m$ is a vector-valued mapping. Given the current iterate $x^k$, LM updates the point by $x^{k+1} = x^k + p^k$, where the search direction $p^k$ is computed as $p^k = -(J_k^T J_k + \mu_k I)^{-1} J_k^T F(x^k)$, $J_k = J(x^k)$, and $J(x) = \nabla_x F(x)$ is the Jacobian matrix of $F(x)$. Various works have extended LM to the case of non-smooth functions (Facchinei & Kanzow, 1997; Jiang, 1999; Jolaoso et al., 2023), where the Jacobian matrix $J(x^k)$ is replaced by its generalized version. Although some LM variants are able to handle non-smooth and singular problems with appealing convergence properties (Du & Gao, 2011; Ueda & Yamashita, 2012; Qi et al., 2016), we point out that one major difference between LM and Newton-type methods is the form of the linear system that defines the search direction $p^k$. For LM, $p^k = -(V_k^T V_k + \mu_k I)^{-1} V_k^T g_k$, whereas the Newton direction is $p^k = -(V_k + \mu_k I)^{-1} g_k$. Clearly, LM almost squares the condition number of the matrix to be inverted when $\mu_k$ is small, which may result in numerical instability and drastically slow down iterative linear solvers such as the conjugate gradient method. Therefore, when $V_k$ is known to be symmetric positive semi-definite, Newton-type methods are typically preferred to LM methods.

**Computational OT** Efficient computation of OT has long been an active research topic in machine learning. In its original form, OT can be solved via linear programming in $\mathcal{O}(n^3 \log n)$ time (Pele & Werman, 2009), where $n$ is the number of data points. The seminal work Cuturi (2013) introduces the entropic-regularized OT and the well-known Sinkhorn algorithm, and numerous works have been proposed to improve it (Altschuler et al., 2017; Dvurechensky et al., 2018; Schmitzer, 2019; Guminov et al., 2021). For sparse OT, there are also a number of specialized algorithms developed to solve the QROT problem (Lorenz et al., 2021; Pasechnyuk et al., 2023).

## 5 NUMERICAL EXPERIMENTS

In this section we test the performance of S5N on a large collection of sparse OT problems, and compare it with various specialized algorithms for QROT. We roughly categorize the existing algorithms into three groups:

- **Group I**: the classical gradient descent method (GD) and other first-order and coordinate descent methods summarized in Pasechnyuk et al. (2023), including the adaptive primal-dual accelerated gradient descent (APDAGD), the block coordinate descent (BCD), and the primal-dual accelerated alternating minimization (PDAAM).
- **Group II**: quasi-Newton methods, including two L-BFGS methods applied to the dual and semi-dual objective function, respectively (Blondel et al., 2018).
- **Group III**: Newton-type methods, including the adaptive semi-smooth Newton (ASSN, Xiao et al., 2018) and the globalized and regularized semi-smooth Newton (GRSSN, Lorenz et al., 2021). For GRSSN, it has a shift hyperparameter $\lambda$ similar to the $\lambda_k$ in (2), but in GRSSN, $\lambda$ is fixed and needs to be manually set. We consider various values $\lambda = 0.1, 0.01, 0.00001$.

We then test each algorithm on the following three groups of test problems:

- **Example 1**: OT between two discrete distributions $\nu \in \Delta_n$ and $\mu \in \Delta_m$, where $\nu$ and $\mu$ are the discretizations of two identical continuous distributions.
- **Example 2**: OT between two different discrete distributions $\nu \in \Delta_n$ and $\mu \in \Delta_m$.
- **Image data**: OT between a pair of images from the MNIST data (LeCun et al., 1998) or Fashion-MNIST data (Xiao et al., 2017).

The details of data generation are given in Appendix D.1. Throughout the experiments, we fix the QROT regularization parameter to be $\gamma = 0.1$, and the resulting transport plans are visualized in Appendix D.2. In this section, we show the results of one pair of MNIST images, and the case of $n = m = 512$ for Examples 1 and 2. Results for larger-scale experiments are given in Appendix D.5.

To compare the performance of different algorithms, we plot the dual objective function value against both the iteration number and the actual run time. The results for Example 1 are displayed in Figure 1. From the plots it is clear that all first-order algorithms show a slow convergence, and in particular, PDAAM also has oscillations in the dual objective function values. As for the quasi-Newton algorithms, although the semi-dual L-BFGS has a fast convergence in iteration number, it suffers from a long run time, since its gradient computation is more difficult and time-consuming than other methods. Within the Newton-type algorithms, ASSN has a very slow convergence, and the performance of GRSSN is extremely sensitive to the shift hyperparameter. Overall, S5N shows visible advantages in terms of computational efficiency.

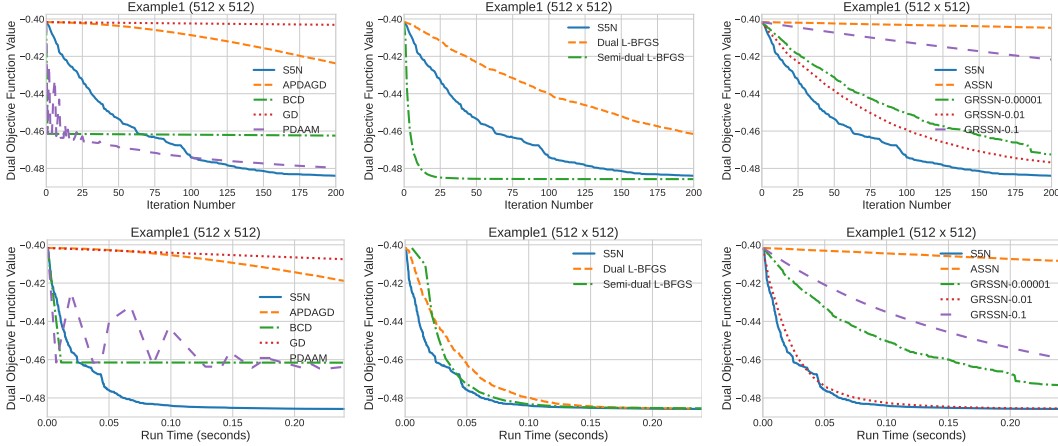

Figure 1: Example 1 ($n = m = 512$). First row: dual objective function value vs. iteration number. Second row: dual objective function value vs. run time.

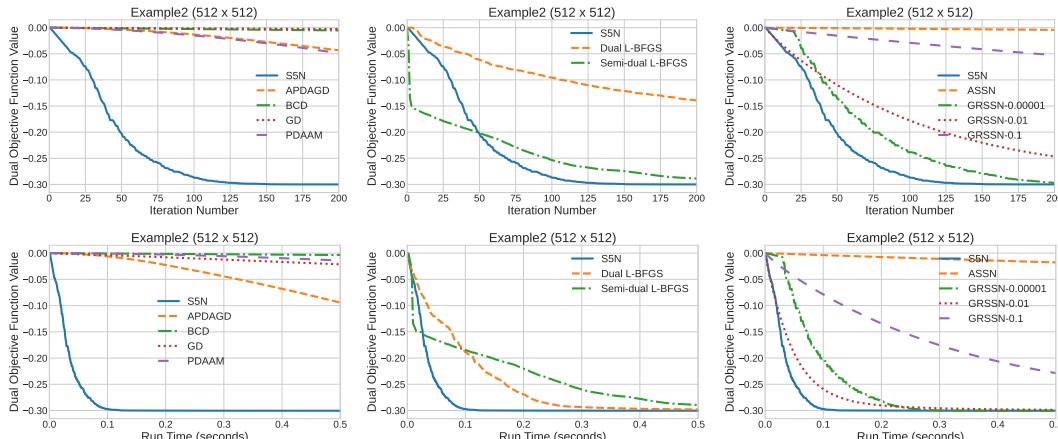

Figure 2: Example 2 ($n = m = 512$). First row: dual objective function value vs. iteration number. Second row: dual objective function value vs. run time.

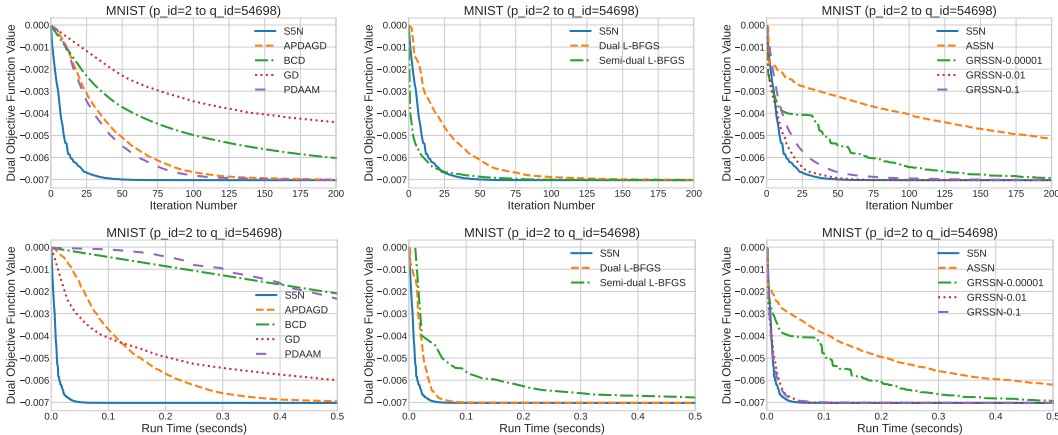

Figure 3: MNIST (image IDs 2-54698). First row: dual objective function value vs. iteration number. Second row: dual objective function value vs. run time.

When it comes to Example 2 (Figure 2), S5N significantly outperforms other algorithms in both iteration count and run time. It is worth noting that in terms of iteration steps, GRSSN performs best with $\lambda = 0.01$ in Example 1, whereas it excels with $\lambda = 0.00001$ in Example 2. This suggests that GRSSN's efficacy heavily relies on the selection of the shift hyperparameter. In contrast, S5N has a self-shifting mechanism that dynamically adjusts $\lambda_k$, demonstrating a high robustness. Example 2 suggests that in scenarios where the two distributions in OT are very different, the advantage of S5N might be even more evident.

Finally, the results for MNIST data as shown in Figure 3 lead to similar conclusions: S5N consistently demonstrates a fast convergence both in iteration number and run time compared with first-order and quasi-Newton methods, and offers the advantage of being adaptive and tuning-free compared with other Newton-type algorithms.

## 6 CONCLUSION

In this article, we propose the S5N algorithm as a novel second-order optimization method for challenging non-smooth and singular problems. We have rigorously shown that the algorithm is globally convergent, and has a fast local quadratic convergence rate. When viewed as a general optimization method, S5N enjoys fast convergence, high efficiency, and great robustness. Meanwhile, it is tuning-free yet flexible. S5N is also especially useful for solving the sparse OT problems, and we have verified its empirical performance on a large collection of QROT test problems. We anticipate that S5N has the potential of further boosting the computation and research of large-scale sparse OT.

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

## A  ADDITIONAL DEFINITIONS

**Definition 3.** *A function $f : A \subset \mathbb{R}^n \to \mathbb{R}^m$ is locally Lipschitz if for each $x_0 \in A$, there exist constants $M > 0$ and $\delta_0 > 0$ such that $\|f(x) - f(x_0)\| \leq M\|x - x_0\|$ whenever $\|x - x_0\| < \delta_0$.*

**Definition 4.** *A set-valued mapping $F$ is said to be upper semi-continuous at $x$, if for any $\varepsilon > 0$ there exists a $\delta > 0$ such that, for all $y$ in $B(x, \delta)$, $F(y) \subseteq F(x) + B(0, \varepsilon)$.*

## B  PROOFS OF THEOREMS

### B.1  TECHNICAL LEMMAS

**Lemma 1.** *Suppose that $f : \mathbb{R}^n \to \mathbb{R}$ is continuously differentiable. Then for any $x \in \mathbb{R}^n$, $|\eta| \leq \tilde{M}$, and $x + \eta p \in \mathbb{R}^n$ with $\|p\| \to 0$, the following statement holds:*

$$\Phi(x, \eta p) \coloneqq f(x + \eta p) - f(x) - g(x)^T(\eta p) = o(\|p\|),$$

*where $g(x)$ represents the gradient of $f$ at $x$.*

*Proof.* By definition, we need to prove that $\Phi(x, \eta p)/\|p\| \to 0$ when $\|p\| \to 0$. By the mean value theorem for multivariate functions, given that $f$ is continuously differentiable, there exists some $\psi \in (0, 1)$ such that for all $x$ and $x + \eta p$,

$$f(x + \eta p) - f(x) = g(x + \psi \eta p)^T(\eta p).$$

Rewriting the equation, we have

$$\Phi(x, \eta p) = g(x + \psi \eta p)^T(\eta p) - g(x)^T(\eta p).$$

To prove that $\Phi(x, \eta p)$ is $o(\|p\|)$, consider the ratio:

$$\left| \frac{\Phi(x, \eta p)}{\|p\|} \right| = \left| \frac{g(x + \psi \eta p)^T(\eta p) - g(x)^T(\eta p)}{\|p\|} \right|.$$

Given the continuity of $g$, we have $g(x + \psi \eta p) \to g(x)$ as $\|p\| \to 0$. Therefore,

$$\left| \frac{g(x + \psi \eta p)^T(\eta p) - g(x)^T(\eta p)}{\|p\|} \right| = \left| \frac{[g(x + \psi \eta p) - g(x)]^T(\eta p)}{\|p\|} \right|$$

$$\leq \frac{\|g(x + \psi \eta p) - g(x)\| \cdot \|\eta p\|}{\|p\|} \leq \frac{\tilde{M} \cdot \|g(x + \psi \eta p) - g(x)\| \cdot \|p\|}{\|p\|}$$

$$= \tilde{M} \cdot \|g(x + \psi \eta p) - g(x)\| \to 0 \quad \text{as } \|p\| \to 0.$$

The proof is complete. $\qquad\square$

**Lemma 2.** *Let Assumption 1 be satisfied. For any $\varepsilon > 0$, there exists a constant $\delta_0 > 0$ such that for all $x \in L(x^0)$ and $\|p\| \leq \delta_0$ satisfying $x + \eta p \in L(x^0)$, we have*

$$\max_{x \in L(x^0)} |\Phi(x, \eta p)| \leq \frac{\varepsilon \tilde{M}}{4\theta} \|p\|.$$

*Proof.* According to Lemma 1, for any $x \in L(x^0)$ and $x + \eta p \in L(x^0)$ with $\|p\| \to 0$, we have

$$|\Phi(x, \eta p)| = o(\|p\|).$$

Therefore, for any $\varepsilon > 0$ and $x \in L(x^0)$, there exists $\delta_x > 0$ such that for all $\|p\| \leq \delta_x$ satisfying $x + \eta p \in L(x^0)$, we have

$$|\Phi(x, \eta p)| \leq \frac{\varepsilon \tilde{M}}{4\theta} \|p\|.$$

Since

$$L(x^0) \subset \bigcup_{x \in L(x^0)} \mathcal{B}(x, \delta_x),$$

where

$$\mathcal{B}(x, \delta_x) = \{x' \in \mathbb{R}^n : \|x' - x\| < \delta_x\},$$

we have that $\{\mathcal{B}(x, \delta_x) : x \in L(x^0)\}$ is an open covering for $L(x^0)$. Since $L(x^0)$ is non-empty and compact, there exist a finite number of points in $L(x^0)$, say $z_1, z_2, ..., z_r$, such that $z_j \in L(x^0)$ for $j = 1, 2, ..., r$, and

$$L(x^0) \subset \bigcup_{j=1}^{r} \mathcal{B}(z_j, \delta_{z_j}).$$

Let $\delta_0 = \min_{1 \leq j \leq r} \delta_{z_j}$. Then we have that for all $\|p\| \leq \delta_0$ and $x \in L(x^0)$,

$$\max_{x \in L(x^0)} |\Phi(x, \eta p)| \leq \frac{\varepsilon \tilde{M}}{4\theta} \|p\|.$$

The proof is complete. $\qquad\square$

**Lemma 3.** *Let $\{x^k\}$ be generated by Algorithm 1, and let $f_k = f(x^k)$, $g_k = \nabla f(x^k)$, $V_k \in \partial g(x^k)$, and $p^k = -[V_k + \mu_k \|g_k\|^\delta I]^{-1} g_k$. Then we have the lower bound of the predicted reduction:*

$$m_k(0) - m_k(\eta^k p^k) \geq \frac{1}{2} \|g_k\| \cdot \min \left\{ \|\eta^k p^k\|, \frac{\|g_k\|}{\|V_k\|} \right\}.$$

*Proof.* Let $r_k = \|\eta^k p^k\|$. If we take $(p, \lambda) = (\eta^k p^k, \mu_k \|g_k\|^\delta)$, then we can verify that $(p, \lambda)$ meets the following relations:

$$\begin{cases} g_k + V_k p + \lambda p = 0 \\ \lambda \geq 0 \\ r_k - \|p\| \geq 0 \\ \lambda \cdot (r_k - \|p\|) = 0 \end{cases}$$

As $m_k(\cdot)$ is a convex function, we have that $\eta^k p^k$ is a KKT point and an optimal solution to the constrained optimization problem

$$\min_{p \in \mathbb{R}^d} \quad m_k(p) = f_k + g_k^T p + \frac{1}{2} p^T V_k p \tag{5}$$
$$\text{s.t.} \quad \|p\| \leq r_k.$$

Next, consider the Cauchy point defined in Chapter 4 of Jorge & Stephen (2006),

$$p_c^k = -\tau_k \frac{r_k}{\|g_k\|} g_k,$$

where

$$\tau_k = \begin{cases} 1, & \text{if } g_k^T V_k g_k \leq 0 \\ \min \left\{ 1, \frac{\|g_k\|^3}{r_k g_k^T V_k g_k} \right\}, & \text{if } g_k^T V_k g_k > 0 \end{cases}.$$

Then we can obtain the following inequality by Lemma 4.3 of Jorge & Stephen (2006):

$$m_k(0) - m_k(p_c^k) \geq \frac{1}{2}\|g_k\| \cdot \min\left\{r_k, \frac{\|g_k\|}{\|V_k\|}\right\}.$$

Meanwhile, the Cauchy point $p_c^k$ is a feasible point for the constrained problem (5), and $\eta^k p^k$ is an optimal solution for this problem. Therefore, $m_k(p_c^k) \geq m_k(\eta^k p^k)$, and we have

$$m_k(0) - m_k(\eta^k p^k) \geq \frac{1}{2}\|g_k\| \cdot \min\left\{r_k, \frac{\|g_k\|}{\|V_k\|}\right\}.$$

The proof is complete. $\qquad\square$

**Lemma 4.** *Let $g_k$ and $\mu_k$ be generated by Algorithm 1. Under the conditions of Assumption 1 and Assumption 2, if there exists a constant $\varepsilon > 0$ and an integer $K$ such that*

$$\|g_k\| \geq \varepsilon, \qquad \forall k \geq K,$$

*then there must exist a sufficiently large constant $\bar{\mu} > 0$, such that $\mu_{k+1} \leq 4\bar{\mu}$ for all $k \geq K$.*

*Proof.* For a given $\bar{\mu}$, define $I_1 = \{k : k \geq K, \mu_k < \bar{\mu}\}$ and $I_2 = \{k : k \geq K, \mu_k \geq \bar{\mu}\}$. If for some $\bar{\mu}$, $I_2$ is finite, then it trivially holds that all $\mu_k$ has a global upper bound, which leads to the desired conclusion. Therefore, we only consider the case that $I_2$ is infinite for all $\bar{\mu} > 0$. Since $\bar{\mu}$ can be chosen arbitrarily, $\{\mu_k\}_{k \in I_2}$ must be unbounded.

Then we will estimate the bound of the following quantity,

$$|\rho_k - 1| = \left|\frac{m_k(p^k) - f(x^k + p^k)}{m_k(0) - m_k(p^k)}\right|,$$

where $\rho_k$, $m_k$ and $f_k$ are defined in the same manner as Algorithm 1.

According to Assumption 2, one has $\|g_k\| \geq \varepsilon$ and $\|g_k\| \leq \beta_1$ for $k \geq K$. Due to the convexity of $f$, we have that $V_k \in \partial g(x^k)$ is positive semi-definite, which implies that the minimum eigenvalue of the matrix $V_k + \mu_k\|g_k\|^\delta I$ is at least $\mu_k \varepsilon^\delta$. In other words,

$$\|(V_k + \mu_k\|g_k\|^\delta I)^{-1}\| = [\lambda_{\min}(V_k + \mu_k\|g_k\|^\delta I)]^{-1} \leq \mu_k^{-1}\varepsilon^{-\delta}.$$

From Algorithm 1, we have

$$\eta^k p^k = \eta^k(V_k + \mu_k\|g_k\|^\delta I)^{-1}g_k,$$

and hence for all $k \in I_2$,

$$\begin{aligned}
\|\eta^k p^k\| &\leq \eta^k \cdot \|(V_k + \mu_k\|g_k\|^\delta I)^{-1}\| \cdot \|g_k\| \\
&\leq \tilde{M}\mu_k^{-1}\varepsilon^{-\delta} \cdot \beta_1 \leq \tilde{M}\bar{\mu}^{-1}\varepsilon^{-\delta} \cdot \beta_1.
\end{aligned} \tag{6}$$

Equation (6) indicates that $\|\eta^k p^k\|$, $k \in I_2$ can be made arbitrarily small with a sufficiently large $\bar{\mu}$. Therefore, we can choose some $\bar{\mu}$ such that

$$\|\eta^k p^k\| \leq \min\left\{\delta_0, \frac{\varepsilon}{\beta_2}, \frac{\varepsilon}{2\theta\beta_2}\right\}, \quad \forall k \in I_2,$$

where $\delta_0$ is defined in Lemma 2. Meanwhile, we know

$$\begin{aligned}
\left|f(x^k + \eta^k p^k) - m_k(\eta^k p^k)\right| &= \left|\Phi(x^k, \eta^k p^k) - \frac{1}{2}(\eta^k p^k)^T V_k(\eta^k p^k)\right| \\
&\leq \left|\Phi(x^k, \eta^k p^k)\right| + \frac{1}{2}\|V_k(\eta^k p^k)\| \cdot \|\eta^k p^k\|.
\end{aligned}$$

Based on Lemma 2, when $\|\eta^k p^k\| \leq \delta_0$, the function $\Phi$ satisfies

$$\left|\Phi(x^k, \eta^k p^k)\right| \leq \frac{\varepsilon\tilde{M}}{4\theta} \cdot \|p^k\|.$$

Furthermore,

$$
\begin{aligned}
\frac{1}{2}\|V_k(\eta^k p^k)\| \cdot \|\eta^k p^k\| &\leq \frac{1}{2}(\beta_2 \cdot \|\eta^k p^k\|) \cdot \|\eta^k p^k\| \\
&\leq \frac{1}{2}\tilde{M}\left(\beta_2 \cdot \frac{\varepsilon}{2\theta\beta_2}\right) \cdot \|p^k\| \\
&= \frac{\varepsilon\tilde{M}}{4\theta} \cdot \|p^k\|.
\end{aligned}
$$

In summary, we obtain

$$
\left|f(x^k + \eta^k p^k) - m_k(\eta^k p^k)\right| \leq \left|\Phi(x^k, \eta^k p^k)\right| + \frac{1}{2}\|V_k(\eta^k p^k)\| \cdot \|\eta^k p^k\| \leq \frac{\varepsilon\tilde{M}}{2\theta} \cdot \|p^k\|. \quad (7)
$$

In view of Lemma 3 and $\eta^k \in [\tilde{m}, \tilde{M}]$, for all $k \in I_2$, the lower bound follows,

$$
\begin{aligned}
m_k(0) - m_k(\eta^k p^k) &\geq \frac{1}{2}\|g_k\| \min\left\{\|\eta^k p^k\|, \frac{\|g_k\|}{\|V_k\|}\right\} \\
&\geq \frac{1}{2}\varepsilon \min\left\{\|\eta^k p^k\|, \frac{\varepsilon}{\beta_2}\right\} \\
&= \frac{1}{2}\varepsilon\|\eta^k p^k\| \geq \frac{\tilde{m}}{2}\varepsilon\|p^k\|. \quad (8)
\end{aligned}
$$

Therefore, by combining (7) and (8), we have

$$
|\rho_k - 1| = \left|\frac{m_k(\eta^k p^k) - f(x^k + \eta^k p^k)}{m_k(0) - m_k(\eta^k p^k)}\right| \leq \frac{\tilde{M}}{\theta\tilde{m}} = \rho_0,
$$

which means that $\rho_k \geq 1 - \rho_0$ for all $k \in I_2$. From Algorithm 1, it follows that for all $k \in I_2$, we have $\mu_{k+1} \leq \mu_k$.

Now we can show that $\mu_{k+1} \leq 4\bar{\mu}$ for all $k \geq K$ by reduction. First, enlarge $\bar{\mu}$ when necessary to ensure that $\bar{\mu} > \mu_K$. Then by Algorithm 1, we must have $\mu_{K+1} \leq 4\bar{\mu}$. Now suppose that $\mu_{l+1} \leq 4\bar{\mu}$ for some $l \geq K$. If $\mu_{l+1} \in I_1$, then clearly $\mu_{l+2} \leq 4\bar{\mu}$ immediately holds. Otherwise, $\mu_{l+1} \in I_2$, so by the argument above, we have $\mu_{l+2} \leq \mu_{l+1} \leq 4\bar{\mu}$. In both cases, the conclusion holds. The proof is complete.

$\square$

**Lemma 5.** *Under the same conditions as in Lemma 4, define*

$$
\mathcal{K} = \left\{k : k \geq K, \rho_k \geq \frac{\tilde{M}}{\theta\tilde{m}}\right\}.
$$

*Then we have that $\mathcal{K}$ is finite.*

*Proof.* We use proof by contradiction. Suppose that $\mathcal{K}$ is infinite. Since

$$
\rho_k = \frac{f(x^k) - f(x^k + \eta^k p^k)}{m_k(0) - m_k(\eta^k p^k)},
$$

and note that $x^{k+1} = x^k$ if $\rho_k < 0$, we have $f(x^{k+1}) \leq f(x^k + \eta^k p^k)$. Therefore,

$$
f(x^k) - f(x^{k+1}) \geq \frac{\tilde{M}}{\theta\tilde{m}}(m_k(0) - m_k(\eta^k p^k)) \geq \frac{\tilde{M}}{2\theta\tilde{m}}\varepsilon \min\left\{\|\eta^k p^k\|, \frac{\varepsilon}{\beta_2}\right\}
$$

for all $k \in \mathcal{K}$, where the second inequality is due to Lemma 3. Based on Assumption 1 and Algorithm 1, $f(x^k)$ is monotonically non-increasing with a lower bound, so $f(x^k)$ has a limit, and

$$
\lim_{k \in \mathcal{K}, k \to \infty} \|\eta^k p^k\| = 0.
$$

On the other hand, $p^k = (V_k + \mu_k \|g_k\|^\delta I)^{-1} g_k$, which means that $g_k = (V_k + \mu_k \|g_k\|^\delta I)(\eta^k p^k)$. Then we can show that

$$
\begin{aligned}
\varepsilon \leq \|g_k\| &= \|(V_k + \mu_k \|g_k\|^\delta I) p^k\| \\
&\leq \|V_k + \mu_k \|g_k\|^\delta I\| \|p^k\| \\
&\leq [\|V_k\| + \|(\mu_k \|g_k\|^\delta I)\|] \cdot \|p^k\| \\
&\leq (\beta_2 + \mu_k \beta_1^\delta) \|p^k\| \\
&\leq (\beta_2 + \mu_k \beta_1^\delta) \cdot \left\| \frac{\eta^k}{\tilde{m}} \cdot p^k \right\| \\
&= (\beta_2 + \mu_k \beta_1^\delta) \cdot \frac{1}{\tilde{m}} \cdot \|\eta^k p^k\|
\end{aligned}
$$

In other words,

$$
0 \leq \frac{\varepsilon \tilde{m}}{\beta_2 + \mu_k \beta_1^\delta} \leq \|\eta^k p^k\| \to 0,
$$

which implies that $\mu_k \to \infty$ for $k \in \mathcal{K}$, $k \to \infty$. This contradicts with the fact that $\mu_k \leq 4\bar{\mu}$ as shown in Lemma 4. The proof is complete. $\qquad\square$

**Lemma 6** (Zhou & Toh, 2005). *Let $G \in \mathbb{R}^{n \times n}$ be a positive semidefinite matirx and $\lambda_0 > 0$ such that*

$$
\|(G + \lambda_0)^{-1}\| \leq \frac{1}{\lambda_0},
$$

*and*

$$
\|(G + \lambda_0 I)^{-1} G\| \leq 2.
$$

*Proof.* Let $\sigma_{min}$ be the smallest singular value of $G + \lambda_0 I$ and $\lambda_{min}$ be the smallest eigenvalues of $\frac{1}{2}(G + G^T) + \lambda_0 I$. By Horn & Johnson (1991), Corollary 3.1.5.

$$
\sigma_{min} \geq \lambda_{min}.
$$

Since $G$ is a positive semi-definite, we have

$$
\begin{aligned}
\lambda_{min} &= \min \left\{ x^T [\frac{1}{2}(G + G^T) + \lambda_0 I] x \right\} \\
&= \min \left\{ x^T G x + \lambda_0 \right\} \\
&\geq \lambda_0,
\end{aligned}
$$

where $\|x\| = 1$ and $x \in \mathbb{R}^n$. Hence we can obtain

$$
\|(G + \lambda_0 I)^{-1}\| = \frac{1}{\sigma_{min}} \leq \frac{1}{\lambda_{min}} \leq \frac{1}{\lambda_0}.
$$

Therefore

$$
\begin{aligned}
\|(G + \lambda_0 I)^{-1} G\| &= \|(G + \lambda_0 I)^{-1}(G + \lambda_0 I - \lambda_0 I)\| \\
&= \|I - \lambda_0 (G + \lambda_0 I)^{-1}\| \\
&\leq 1 + \lambda_0 \|(G + \lambda_0 I)^{-1}\| \\
&\leq 2.
\end{aligned}
$$

The proof is complete. $\qquad\square$

**Lemma 7.** *Under the conditions of Assumption 3, when $x^k \in N(x^*, r_1/2)$, there is a constant $c_2 > 0$ such that*

$$
\|\eta^k p^k\| \leq c_2 \cdot \mathrm{dist}(x^k, \mathcal{X}).
$$

*Proof.* Since $x^k \in N(x^*, r_1/2)$, we have

$$
\|\bar{x}^k - x^*\| \leq \|\bar{x}^k - x^k\| + \|x^k - x^*\| \leq \|x^* - x^k\| + \|x^k - x^*\| \leq r_1,
$$

which means that

$$\bar{x}^k \in N(x^*, r_1). \tag{9}$$

Let $\lambda_k = \mu_k \|g_k\|^\delta$, and we know from Assumption 3,

$$\lambda_k = \mu_k \|g_k\|^\delta \geq \kappa c_1^\delta \cdot \operatorname{dist}(x^k, \mathcal{X})^\delta = \kappa c_1^\delta \|\bar{x}^k - x^k\|^\delta.$$

Based on Lemma 6, we have

$$
\begin{aligned}
\|\eta^k p^k\| &= \|\eta^k (V_k + \lambda_k I)^{-1} g_k\| \\
&\leq \tilde{M} \left\{ \|(V_k + \lambda_k I)^{-1}[g_k - g(\bar{x}^k) - V_k(\bar{x}^k - x^k)]\| + \|(V_k + \lambda_k I)^{-1} V_k(\bar{x}^k - x^k)\| \right\} \\
&\leq \tilde{M} \left\{ \lambda_k^{-1} \cdot L \|\bar{x}^k - x^k\|^2 + 2\|\bar{x}^k - x^k\| \right\}
\end{aligned}
$$

Since $\|g(x)\|$ is bounded from Assumption 2. Thus,

$$
\begin{aligned}
\lambda_k^{-1} \|\bar{x}^k - x^k\| &= \frac{\operatorname{dist}(x^k, \mathcal{X})}{\mu_k \|g_k\|^\delta} \\
&\leq \frac{\|g_k\|^{1-\delta}}{c_1 \kappa} \\
&\leq \frac{\beta_1^{1-\delta}}{c_1 \kappa}
\end{aligned}
$$

Then we have

$$
\begin{aligned}
\|\eta^k p^k\| &= \|\eta^k (V_k + \lambda_k I)^{-1} g_k\| \\
&\leq \tilde{M} \left\{ \lambda_k^{-1} \cdot L \|\bar{x}^k - x^k\|^2 + 2\|\bar{x}^k - x^k\| \right\} \\
&\leq \tilde{M} \left\{ \frac{\beta_1^{1-\delta}}{c_1 \kappa} L \|\bar{x}^k - x^k\| + 2\|\bar{x}^k - x^k\| \right\} \\
&= \tilde{M} \left( \frac{\beta_1^{1-\delta} L}{c_1 \kappa} + 2 \right) \|\bar{x}^k - x^k\| = c_2 \cdot \operatorname{dist}(x^k, \mathcal{X}).
\end{aligned}
$$

$\square$

**Lemma 8.** *Suppose that $f : \mathbb{R}^n \to \mathbb{R}$ is a $LC^1$ function (i.e., $f$ is differentiable and its derivative $g$ is locally Lipschitz; see Qi, 1994). Then for any $x \in \mathbb{R}^n$ and $x + \eta p \in \mathbb{R}^n$ with $\|p\| \to 0$, the following statements hold true.*

$$f(x + \eta p) - f(x) - g^T(\eta p) - \frac{1}{2}(\eta p)^T V(\eta p) = o(\|p\|^2)$$

*Proof.* According to second-order mean value theorem Hiriart-Urruty et al. (1984)(Theorem 2.3) for $LC^1$ functions, for any $x, x + p \in \mathbb{R}^n$ and $V \in \partial g(x)$, there exists $t \in [0, 1]$ and $V' \in \partial g(x + t\eta p)$, where $V'$ is the projection of $V$ on set $\partial g(x + t\eta p)$, such that

$$f(x + \eta p) - f(x) - g^T(\eta p) = \frac{1}{2} \eta p^T V' \eta p$$

From Theorem 2 and the upper semicontinuiy of $\partial g(x)$, we, as $\|p\| \to 0$, $V' \to V$, which means,

$$
\begin{aligned}
\left| \frac{f(x + \eta p) - f(x) - g^T(\eta p) - \frac{1}{2}(\eta p)^T V(\eta p)}{\|p\|^2} \right| &= \left| \frac{\frac{1}{2}(\eta p)^T V'(\eta p) - \frac{1}{2}(\eta p)^T V(\eta p)}{\|p\|^2} \right| \\
&\leq \frac{\tilde{M}^2}{2} \left| \frac{\|V' - V\| \cdot \|p\|^2}{\|p\|^2} \right| \\
&= \frac{\tilde{M}^2}{2} \|V' - V\| \\
&\to 0 \quad (\|p\| \to 0)
\end{aligned}
$$

$\square$

**Lemma 9.** *Under Assumption 1 to Assumption 3, when $x^k \in N(x^*, r_1/2)$ there exists a constant $\bar{\kappa} > \kappa$ such that*

$$\mu_k \leq \bar{\kappa}$$

*Proof.* By Assumption 3 and Lemma 7, when $x^k \in N(x^*, r_1/2)$,

$$
\begin{aligned}
m_k(0) - m_k(\eta^k p^k) &\geq \frac{1}{2}\|g_k\| \min\left\{\|\eta^k p^k\|, \frac{\|g_k\|}{\|V_k\|}\right\} \\
&\geq \frac{c_1}{2}\|x^k - \bar{x}^k\| \min\left\{\|\eta^k p^k\|, \frac{c_1}{\beta_2}\|x^k - \bar{x}^k\|\right\} \\
&\geq \frac{c_1}{2c_2}\|\eta^k p^k\| \min\left\{\|\eta^k p^k\|, \frac{c_1}{\beta_2 c_2}\|\eta^k p^k\|\right\}
\end{aligned}
$$

Based on Lemma 8, for $\|p^k\| \to 0$, we can obtain,

$$
\begin{aligned}
|\rho_k - 1| &= \left| \frac{f(x^k + \eta^k p^k) - m_k(\eta^k p^k)}{m_k(0) - m_k(\eta^k p^k)} \right| \\
&\leq \frac{o(\|p^k\|^2)}{\frac{c_1}{2c_2}\|\eta^k p^k\| \min\left\{\|\eta^k p^k\|, \frac{c_1}{\beta_2 c_2}\|\eta^k p^k\|\right\}} \to 0.
\end{aligned}
$$

Therefore, from Lemma 7 we can obain that when $\operatorname{dist}(x^k, \mathcal{X}) \to 0$ so that $\|\eta^k p^k\| \to 0$, and then $\rho_k \to 1$. Hence, we have that there exists a constant $\bar{\kappa} > \kappa > 0$ such that $\mu_k \leq \bar{\kappa}$. $\qquad\square$

**Lemma 10.** $\{x^k\}$ *is generated by Algorithm 1. If $x^k, x^{k+1} \in N(x^*, r_1/2)$, and then there exists a constants $c_3 > 0$ such that*

$$\operatorname{dist}(x^k + \eta^k p^k, \mathcal{X}) \leq c_3 \cdot \operatorname{dist}(x^k, \mathcal{X})^2.$$

*Proof.* According to Assumption 3, $g(x)$ is Lipschitz continuous on $N(x^*, r_1/2)$, and then there exists a constant $\tilde{L}$ such that

$$\|g(y) - g(x)\| \leq \tilde{L}\|y - x\|, \qquad \forall x, y \in N(x^*, r_1/2).$$

From Lemma 7 and Lemma 9, there exist constants $K_1$ and $K_2$ such that

$$K_1\|\bar{x}^k - x^k\|^\delta \leq \lambda_k = \mu_k\|g_k\|^\delta = \mu_k\|g(\bar{x}^k) - g_k\|^\delta \leq K_2\|\bar{x}^k - x^k\|^\delta,$$

where $K_1 = c_1^\delta \kappa$ and $K_2 = \tilde{L}^\delta \bar{\kappa}$.

Since $\rho_k \to 1$ which means $\rho_k > 0$ for all $k$ large enough, according to Algorithm 1 we have

$$x^{k+1} = x^k + \eta^k p^k.$$

From the conditions with Assumptions 3, we can obtain for $\delta \in [1, 2]$

$$
\begin{aligned}
c_1\|\bar{x}^{k+1} - x^{k+1}\| &\leq \|g_{k+1}\| \\
&= \|g(x^k + \eta^k p^k)\| \\
&\leq \|g(x^k + \eta^k p^k) - g(x^k) - V_k(\eta^k p^k)\| + \|g(x^k) + V_k(\eta^k p^k)\| \\
&\leq L\|\eta^k p^k\|^2 + \|\lambda_k I \cdot p^k\| \\
&\leq L\|\eta^k p^k\|^2 + \frac{\lambda_k}{\tilde{m}}\|\eta^k p^k\| \\
&\leq L c_2^2 \cdot \operatorname{dist}(x^k, \mathcal{X})^2 + \frac{K_2 c_2}{\tilde{m}} \cdot \operatorname{dist}(x^k, \mathcal{X})^{1+\delta} \\
&\leq \left(L c_2^2 + \frac{K_2 c_2}{\tilde{m}}\right) \operatorname{dist}(x^k, \mathcal{X})^2
\end{aligned}
$$

which means

$$\operatorname{dist}(x^{k+1}, \mathcal{X}) \leq \left(\frac{L c_2^2}{c_1} + \frac{K_2 c_2}{c_1 \tilde{m}}\right) \operatorname{dist}(x^k, \mathcal{X})^2.$$

The proof is complete. $\qquad\square$

**Lemma 11.** *Let $r = \min\left\{\frac{r_1}{2+4c_2}, \frac{1}{2c_3}\right\}$. And then there exists a integer $K$ large enough such that when $x^K \in N(x^*, r)$, then we can obtain that $x^k \in N(x^*, r_1/2)$ for all $k \geq K$.*

*Proof.* Let $\bar{x}^i$ be the projection of $x^i$ onto $\mathcal{X}$. We define $i$ as the $i$-th item following $K$, where $i$ takes the values $1, \ldots, k$, which means that $x^K$ represents the item when $i = 0$.

And then we first consider $i = 1$, which corresponds to the $x^{K+1}$ term,

$$
\begin{aligned}
|x^1 - x^*\| &= \|x^0 + \eta^0 p^0 - x^*\| \leq \|x^0 - x^*\| + \|\eta^0 p^0\| \\
&\leq \|x^0 - x^*\| + c_2\|x^0 - \bar{x}^0\| \leq (1 + c_2)r \leq r_1/2,
\end{aligned}
$$

which means $x^1 \in N(x^*, r_1/2)$. Suppose $x^i \in N(x^*, r_1/2)$ for $i = 2, \ldots, k$. Then we have from Lemma 10 that

$$
\begin{aligned}
\|x^i - \bar{x}^i\| &\leq c_3 \|x^{i-1} - \bar{x}^{i-1}\|^2 \\
&\leq \cdots \leq c_3^{2^i - 1} \|x^0 - x^*\|^{2^i} \\
&\leq r\left(\frac{1}{2}\right)^{2^i - 1}.
\end{aligned}
$$

Hence,

$$
\begin{aligned}
\|x^{k+1} - x^*\| &\leq \|x^1 - x^*\| + \sum_{i=1}^{k} \|\eta^i p^i\| \\
&\leq (1 + c_2)r + c_2 \sum_{i=1}^{k} \|x^i - \bar{x}^i\| \\
&\leq (1 + c_2)r + c_2 r \sum_{i=1}^{k} \left(\frac{1}{2}\right)^{2^i - 1} \\
&\leq (1 + c_2)r + c_2 r \sum_{i=1}^{\infty} \left(\frac{1}{2}\right)^{i} \\
&\leq (1 + 2c_2)r \\
&\leq r_1/2.
\end{aligned}
$$

where the last inequality follows from $r \leq \frac{r_1}{2+4c_2}$. Consequently we have $x^{k+1} \in N(x^*, r_1/2)$. $\square$

**Lemma 12.** *The derivative of the dual function $\Phi$ is (globally) Lipschitz continuous and semi-smooth.*

*Proof.* Let $f : O \to R^m$. It is defined as piecewise $C^k$ (with $k \in [1, \infty]$), if $f$ remains continuous at each point $\bar{x}$. Moreover, within a neighborhood $V \subset O$ around $\bar{x}$, we have

$$
f(x) \in \{f_1(x), f_2(x), \ldots, f_N(x)\} \quad \forall x \in O,
$$

where each $f_i(x)$ for $i = 1, 2, \ldots, N$ belongs to the class of $C^k$ functions. As demonstrated in Ulbrich (2011), Proposition 2.26, a function being piecewise $C^1$ is semi-smooth, and if it is piecewise $C^2$, it elevates to strongly semi-smooth. $\square$

**Lemma 13.** *Let $f(x)$ be the same as the problem (4),*

$$
f(x) := f(\alpha, \beta) = \frac{1}{2} \left\| (\alpha \oplus \beta - C)_+ \right\|_F^2 - \gamma \langle \nu, \alpha \rangle - \gamma \langle \mu, \beta \rangle
$$

*Then we can deduce its generalized Jacobian*

$$
V = \begin{pmatrix} \mathbf{diag}(\sigma 1_m) & \sigma \\ \sigma^T & \mathbf{diag}(\sigma^T 1_n) \end{pmatrix} \in \partial g(\alpha, \beta),
$$

*where $\sigma_{ij} = 1$ if $\alpha_i + \beta_j - c_{ij} \geq 0$, otherwise, $\sigma_{ij} = 0$*

*Proof.* The proof of this lemma is inspired by Clarke (1990) Example 2.3.17, Theorem 2.3.10 and Hiriart-Urruty et al. (1984) Example 2.1, We first deduce its derivative $g(\alpha, \beta)$,

$$g(\alpha, \beta) = \begin{pmatrix} (\alpha \oplus \beta - C)_+ 1_m - \gamma\nu \\ 1_n^T(\alpha \oplus \beta - C)_+ - \gamma\mu \end{pmatrix} = \begin{pmatrix} \sum_{j=1}^m (\alpha_i + \beta_j - c_{ij})_+ - \gamma\nu_i \\ \sum_{i=1}^n (\alpha_i + \beta_j - c_{ij})_+ - \gamma\mu_j \end{pmatrix}$$

for $i = 1, \ldots, n$ and $j = 1, \ldots, m$. $g(\alpha, \beta)$ can be represented as

$$g(\alpha, \beta) = \begin{pmatrix} g_1(\alpha, \beta) \\ g_2(\alpha, \beta) \\ \ldots \\ g_{n+m}(\alpha, \beta) \end{pmatrix}$$

For $g_i(\alpha, \beta) = \sum_{j=1}^m (\alpha_i + \beta_j - c_{ij})_+ - \gamma\nu_i$, its generalized derivative with respect to $\alpha_i$ is,

$$\frac{\partial g_i(\alpha, \beta)}{\partial \alpha_i} = \begin{cases} 1, & \text{if } \alpha_i + \beta_j - c_{ij} > 0 \\ 0, & \text{if } \alpha_i + \beta_j - c_{ij} < 0 \\ \text{None}, & \text{if } \alpha_i + \beta_j - c_{ij} = 0 \end{cases}$$

Considering sequences $\alpha_i + \beta_j - c_{ij}$ approaching from the right, we have

$$\lim_{(\alpha_i + \beta_j - c_{ij}) \to 0^+} \nabla g_i(\alpha_i + \beta_j - c_{ij}) = 1$$

Similarly, from the left, we have:

$$\lim_{(\alpha_i + \beta_j - c_{ij}) \to 0^-} \nabla g_i(\alpha_i + \beta_j - c_{ij}) = 0$$

According to Definition 1, we are interested in the convex hull of these limiting gradients at $\alpha_i + \beta_j - c_{ij} = 0$. The convex hull of the set $\{0, 1\}$ is the interval $[0, 1]$. Thus, for $g_i(\alpha, \beta)$ at $\alpha_i + \beta_j - c_{ij} = 0$,

$$\partial_C g_i(0) = [0, 1].$$

Thus we can compute $\frac{\partial g(\alpha, \beta)}{\partial \alpha}$ and $\frac{\partial g(\alpha, \beta)}{\partial \beta}$ to get one of the Clarke Jacobian $V \in \partial g(\alpha, \beta)$.

$$V = \begin{pmatrix} \mathbf{diag}(\sigma 1_m) & \sigma \\ \sigma^T & \mathbf{diag}(\sigma^T 1_n) \end{pmatrix} \in \partial g(\alpha, \beta),$$

where $\sigma_{ij} = 1$ if $\alpha_i + \beta_j - c_{ij} \geq 0$, and $\sigma_{ij} = 0$ otherwise. The proof is complete. $\square$

**Lemma 14.** *The Clarke's generalized Jacobian matrix $V$ in Lemma 13 is symmetric and positive semi-definite.*

*Proof.* The proof of this lemma is inspired by Lorenz et al. (2021) Lemma 3.2. From the construction of the Jacobian matrix $V$ in Lemma 13, it can be observed that the upper-right block and the lower-left block of the block matrix are a pair of transposed matrices.

$$V = \begin{pmatrix} \mathbf{diag}(\sigma 1_m) & \sigma \\ \sigma^T & \mathbf{diag}(\sigma^T 1_n) \end{pmatrix} \in \partial g(\alpha, \beta),$$

Hence, this Jacobian matrix is symmetric.
For any vector $a \in \mathbb{R}^n$ and $b \in \mathbb{R}^m$, it follows that

$$(a, b)^T V(a, b) = \sum_{j=1}^m \sum_{i=1}^n \sigma_{ij} a_i^2 + \sum_{j=1}^m \sum_{i=1}^n \sigma_{ij} b_j^2 + 2 \sum_{j=1}^m \sum_{i=1}^n \sigma_{ij} a_i b_j = \sum_{j=1}^m \sum_{i=1}^n \sigma_{ij}(a_i + b_j)^2 \geq 0.$$

which can be infered that the Jacobian matrix $V$ is positive semi-definite, $\square$

## B.2 Proof of Theorem 3

We proceed the proof by contradiction. Suppose that there exist some $\varepsilon > 0$ and an integer $K$ such that

$$\|g_k\| > \varepsilon, \ \forall k \geq K.$$

Then Lemma 5 implies that the index set

$$\mathcal{K} = \left\{ k : k \geq K, \rho_k \geq \frac{\tilde{M}}{\theta \tilde{m}} = \rho_0 \right\}$$

is finite. This means that there is a sufficiently large integer $K'$ such that $\rho_k < \rho_0$ for all $k \geq K'$. According to Algorithm 1, we must have

$$\mu_{k+1} = 4\mu_k, \qquad \forall k \geq K',$$

which means that $\mu_k \to \infty$. However, this contradicts with the fact that $\mu_k \leq 4\bar{\mu}$ for some $\bar{\mu} > 0$ as shown in Lemma 4. The proof is complete.

## B.3 Proof of Theorem 4

From Lemma 11 we have that there exists a sufficiently large integer $K, \forall k > K, x^k \in N(x^*, r_1/2)$ when $x^K \in N(x^*, r)$. According to Lemma 10 we can obtain

$$\text{dist}(x^k + \eta^k p^k, \mathcal{X}) \leq c_3 \cdot \text{dist}(x^k, \mathcal{X})^2.$$

The proof is complete.

## B.4 Proof of Proposition 1

Based on the lemma 12, we have $f(x)$ is a piecewise quadratic function, and its derivative $g(x)$ is a piecewise linear function, thus $g(x)$ is strongly semi-smooth. According to Lemma 13, we can obtain the gradient $g(\alpha, \beta)$. One of the Clarke's generalized Jacobian matrix $V \in \mathbb{R}^{n+m}$ is calculated as:

$$V = \begin{pmatrix} \mathbf{diag}(\sigma 1_m) & \sigma \\ \sigma^T & \mathbf{diag}(\sigma^T 1_n) \end{pmatrix} \in \partial g(\alpha, \beta),$$

where $\sigma_{ij} = 1$ if $\alpha_i + \beta_j - c_{ij} \geq 0$, and $\sigma_{ij} = 0$ otherwise. From Lemma 14, we can deduce that $V$ is a symmetric positive semi-definite matrix.

## C Computing the Newton Directions

We need to solve linear systems of the form $(V + \lambda I)^{-1} x$, where $V \in \mathbb{R}^{(n+m) \times (n+m)}$ is a positive semi-definite matrix, $\lambda > 0$, $x = (w^T, z^T)^T$, $w \in \mathbb{R}^n$, and $z \in \mathbb{R}^m$.

For QROT problems, $V$ has the following form:

$$V = \begin{bmatrix} \mathbf{diag}(h_1) & \sigma \\ \sigma^T & \mathbf{diag}(h_2) \end{bmatrix},$$

where $\sigma \in \mathbb{R}^{n \times m}$ is a sparse matrix, $h_1 = \sigma \mathbf{1}_m$, and $h_2 = \sigma^T \mathbf{1}_n$.

Recall the inverse formula for block matrices:

$$\begin{bmatrix} A & B \\ C & D \end{bmatrix}^{-1} = \begin{bmatrix} A^{-1} + A^{-1} B \left( D - CA^{-1}B \right)^{-1} CA^{-1} & -A^{-1}B \left( D - CA^{-1}B \right)^{-1} \\ - \left( D - CA^{-1}B \right)^{-1} CA^{-1} & \left( D - CA^{-1}B \right)^{-1} \end{bmatrix}$$

$$= \begin{bmatrix} A^{-1} + A^{-1} B \Delta^{-1} CA^{-1} & -A^{-1}B\Delta^{-1} \\ -\Delta^{-1} CA^{-1} & \Delta^{-1} \end{bmatrix}, \qquad \Delta = D - CA^{-1}B,$$

where $A$ and $\Delta$ are invertible. As a result,

$$
\begin{bmatrix} A & B \\ C & D \end{bmatrix}^{-1} \begin{bmatrix} w \\ z \end{bmatrix} = \begin{bmatrix} A^{-1}w + A^{-1}B\Delta^{-1}CA^{-1}w - A^{-1}B\Delta^{-1}z \\ \Delta^{-1}z - \Delta^{-1}CA^{-1}w \end{bmatrix}
$$

$$
= \begin{bmatrix} A^{-1}w + A^{-1}B\Delta^{-1}y - A^{-1}B\Delta^{-1}z \\ \Delta^{-1}z - \Delta^{-1}y \end{bmatrix}
$$

$$
= \begin{bmatrix} v - A^{-1}B\Delta^{-1}(z - y) \\ \Delta^{-1}(z - y) \end{bmatrix}, \qquad v = A^{-1}w, \quad y = Cv.
$$

Therefore, for QROT problems, first compute

$$
v = w \circ (h_1 + \lambda \mathbf{1}_n)^{-1}, \quad y = \sigma^T v,
$$

where $\circ$ is the elementwise multiplication operator, and for a vector $v$, $v^{-1}$ means taking the reciprocal of each element of $v$. Then we have $(V + \lambda I)^{-1}x = (r_1^T, r_2^T)^T$, where

$$
r_2 = \Delta^{-1}(z - y)
$$

$$
r_1 = v - (\sigma r_2) \circ (h_1 + \lambda \mathbf{1}_n)^{-1}.
$$

We use conjugate gradient method to compute $r_2 = \Delta^{-1}(z - y)$, so we need to implement the operator $u \to \Delta u$. Note that $\Delta = \mathbf{diag}(h_2) + \lambda I_m - \sigma^T[\mathbf{diag}(h_2) + \lambda I_n]^{-1}\sigma$, so

$$
\Delta u = h_2 \circ u + \lambda u - \sigma^T[(\sigma u) \circ (h_1 + \lambda \mathbf{1}_n)^{-1}].
$$

## D  ADDITIONAL EXPERIMENT RESULTS

### D.1  TEST PROBLEMS

We consider four groups of test problems:

- **Example 1**: OT between the discretizations of two identical distributions. We set $\nu = n^{-1}\mathbf{1}_n$, $\mu = m^{-1}\mathbf{1}_m$, and the cost matrix $C_{ij} = \|X_i - Y_j\|^2$, where $X_i \sim N(0, I_{10})$ and $Y_j \sim N(0, I_{10})$.
- **Example 2**: OT between two different distributions. Let $x_i = 5(i-1)/(n-1)$, $i = 1, \ldots, n$, and $y_j = 5(j-1)/(m-1)$, $j = 1, \ldots, m$, which are equally-spaced points on $[0, 5]$. Define the cost matrix as $C_{ij} = (x_i - y_j)^2$. Let $f_1$ be the density function of an exponential distribution with mean one, and $f_2$ be the density function of a normal mixture distribution $0.2 \cdot N(1, 0.2) + 0.8 \cdot N(3, 0.5)$. Then we set $\tilde{a}_i = f_1(x_i)$, $\tilde{b}_j = f_2(y_j)$, $\nu_i = \frac{\tilde{a}_i}{\sum_{k=1}^n \tilde{a}_k}$, and $\mu_j = \frac{\tilde{b}_j}{\sum_{k=1}^m \tilde{b}_k}$.
- **(Fashion-)MNIST data**: OT between a pair of images from the MNIST data (LeCun et al., 1998) or Fashion-MNIST data (Xiao et al., 2017). The $\nu$ and $\mu$ vectors are the flattened and normalized pixel values, and the cost matrix holds the Euclidean distances between individual pixels.
- **ImageNet data**: OT between two categories of images from the ImageNet data set (Deng et al., 2009). We use a subset of ImageNet from the Imagenette Github repository[2], which contains ten classes of ImageNet images: tench, English springer, cassette player, chain saw, church, French horn, garbage truck, gas pump, golf ball, and parachute. Approximately 1000 images per category are selected. We map each image to a 30-dimensional feature vector by first passing the image to a ResNet18 network, resulting in a 512-dimensional vector, then followed by a dimension reduction by principal component analysis. Let $x_i \in \mathbb{R}^{30}$ be the feature vector of an image in the first category, $i = 1, \ldots, n$, and $y_j \in \mathbb{R}^{30}$ be the feature vector of an image in the second category, $j = 1, \ldots, m$. Then $\nu = n^{-1}\mathbf{1}_n$, $\mu = m^{-1}\mathbf{1}_m$, and the cost matrix is $C_{ij} = \|x_i - y_j\|^2$.

### D.2  VISUALIZATION OF TRANSPORT PLANS

Figures 4 and 5 show the transport plans for both the unregularized OT and QROT, on Example 1 and Example 2. The transport plans for (Fashion-)MNIST and ImageNet data are not easy to visualize due to their large sizes, so they are not included here.

---

[2]https://github.com/fastai/imagenette.

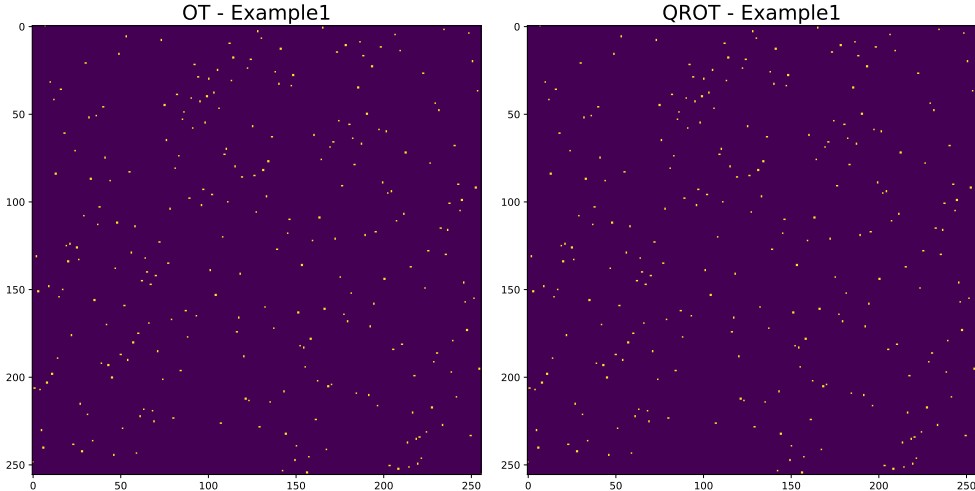

Figure 4: Transport plans for Example 1 ($n = m = 256$): OT vs. QROT.

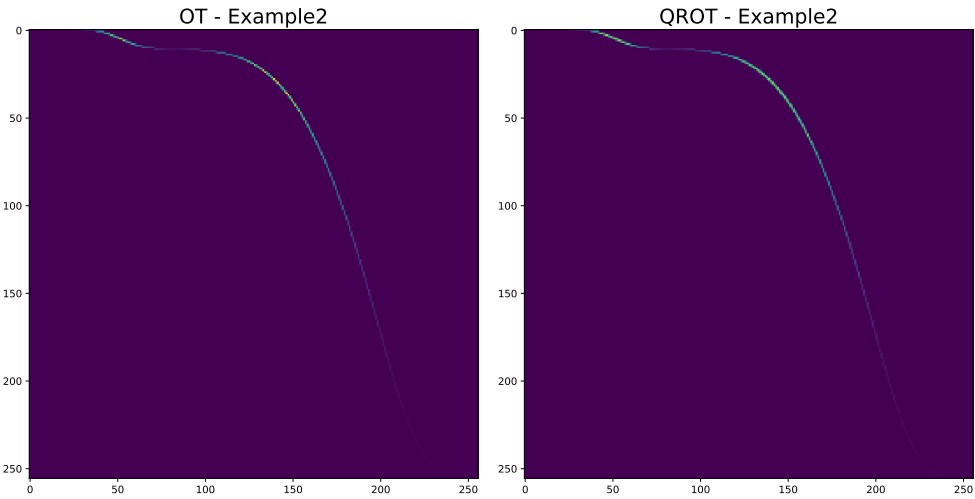

Figure 5: Transport plans for Example 2 ($n = m = 256$): OT vs. QROT.

## D.3 VISUALIZATION OF DUALITY GAPS

Figures 6 to 8 show the convergence of duality gaps of different optimization algorithms.

## D.4 COMPUTING TIME BY ITERATIONS

Tables 2 to 4 show the computing time by iterations for different algorithms on Example 1, Example 2, and MNIST data.

## D.5 ADDITIONAL TEST PROBLEMS

Figures 9 to 20 show additional experiment results as supplements to Section 5. For Example 1 and Example 2, we consider additional problem sizes, $n = m \in \{1024, 2048, 4096\}$. For image data, we show more pairs of images obtained from MNIST or Fashion-MNIST data. We also include results for the ImageNet data in this section.

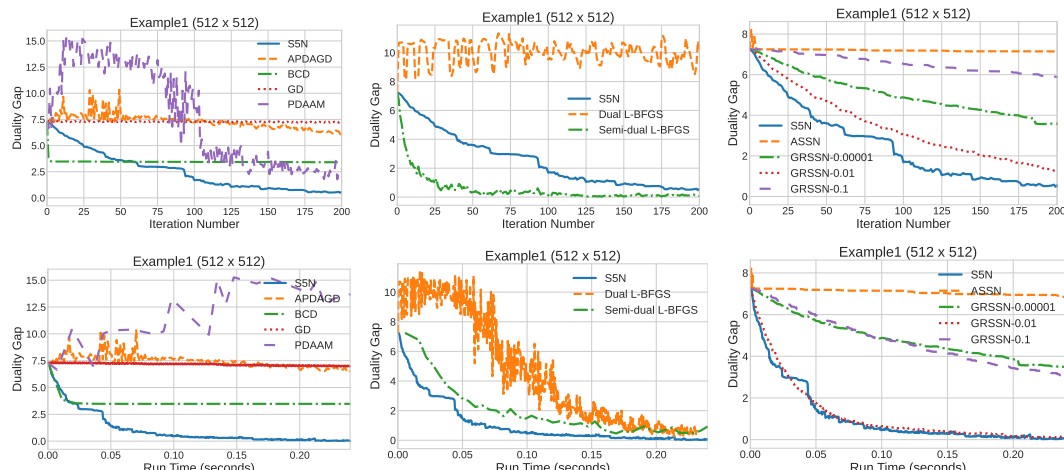

Figure 6: Example 1 ($n = m = 512$). First row: duality gap vs. iteration number. Second row: duality gap vs. run time.

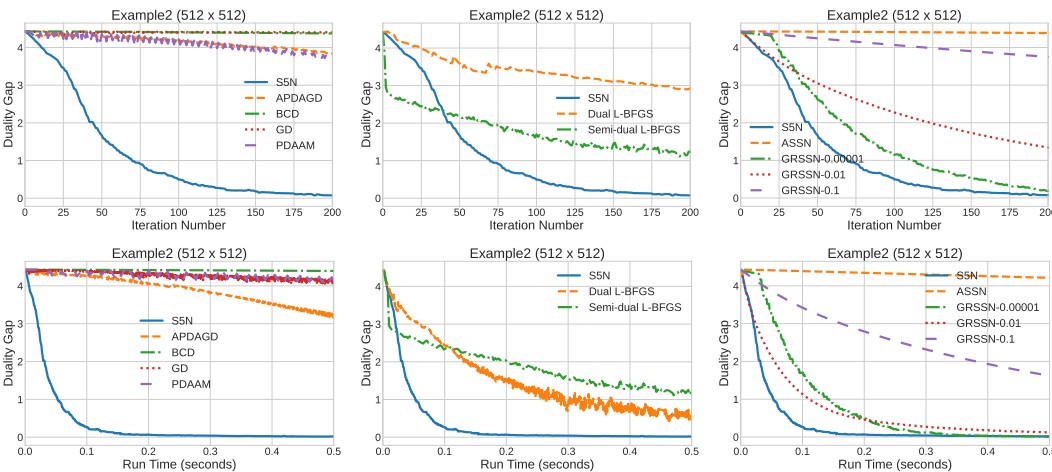

Figure 7: Example 2 ($n = m = 512$). First row: duality gap vs. iteration number. Second row: duality gap vs. run time.

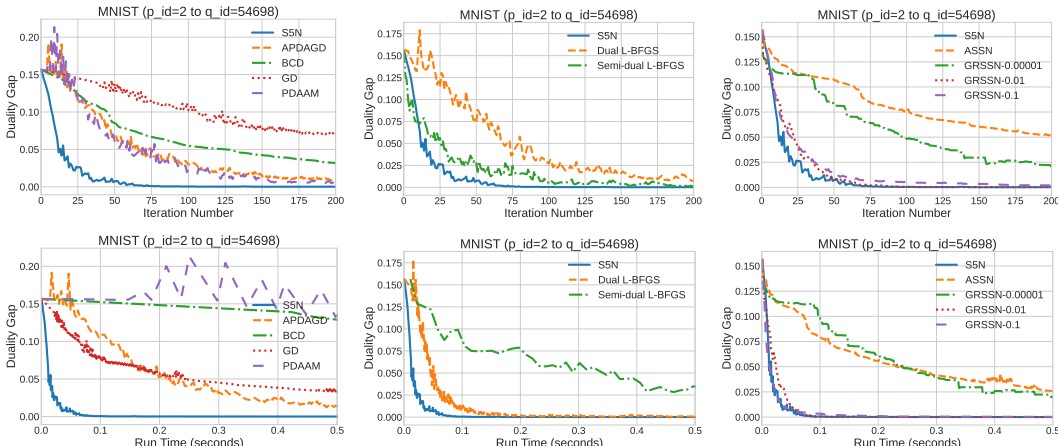

Figure 8: MNIST (image IDs 2-54698). First row: duality gap vs. iteration number. Second row: duality gap vs. run time.

Table 2: Run time (in milliseconds) of algorithms by iterations on Example 1 ($n = m = 512$).

| #Iter | GD | BCD | APDAGD | PDAAM | Dual L-BFGS | Semi-dual L-BFGS | ASSN | GRSSN $\lambda = 0.01$ | S5N |
|-------|------|--------|--------|--------|------|--------|-------|-------|-------|
| 100 | 28.3 | 1071.3 | 138.9 | 1214.0 | 24.0 | 566.5 | 52.1 | 28.2 | 45.3 |
| 200 | 57.0 | 2164.2 | 279.3 | 2428.9 | 48.7 | 1188.6 | 105.6 | 59.3 | 97.9 |
| 300 | 85.8 | 3255.9 | 419.9 | 3647.4 | 71.0 | | 158.9 | 92.5 | 137.4 |
| 400 | 114.5 | 4348.0 | 560.2 | 4859.2 | 93.0 | | 212.4 | 127.1 | 179.4 |
| 500 | 143.3 | 5439.8 | 701.1 | 6072.0 | 118.5 | | 265.7 | 161.2 | 224.4 |

Table 3: Run time (in milliseconds) of algorithms by iterations on Example 2 ($n = m = 512$).

| #Iter | GD | BCD | APDAGD | PDAAM | Dual L-BFGS | Semi-dual L-BFGS | ASSN | GRSSN $\lambda = 0.01$ | S5N |
|-------|-------|--------|--------|--------|-------|--------|-------|--------|-------|
| 100 | 31.3 | 362.9 | 145.4 | 507.1 | 32.6 | 276.4 | 54.02 | 43.7 | 76.4 |
| 200 | 63.4 | 740.4 | 292.7 | 1020.2 | 66.9 | 481.6 | 109.2 | 84.93 | 159.2 |
| 300 | 95.62 | 1120.1 | 440.0 | 1523.0 | 102.3 | 694.9 | 164.4 | 125.6 | 229.0 |
| 400 | 127.5 | 1501.3 | 586.7 | 2010.5 | 134.0 | 908.8 | 219.8 | 166.5 | 296.3 |
| 500 | 159.6 | 1882.9 | 733.7 | 2517.4 | 158.1 | 1106.7 | 275.2 | 208.6 | 369.7 |

Table 4: Run time (in milliseconds) of algorithms by iterations on MNIST data.

| #Iter | GD | BCD | APDAGD | PDAAM | Dual L-BFGS | Semi-dual L-BFGS | ASSN | GRSSN $\lambda = 0.01$ | S5N |
|-------|-------|---------|--------|---------|-------|--------|-------|--------|-------|
| 100 | 59.3 | 2193.0 | 311.7 | 2540.5 | 60.0 | 1364.0 | 110.1 | 96.7 | 106.5 |
| 200 | 132.3 | 4438.3 | 626.6 | 5087.8 | 105.1 | 2988.7 | 225.9 | 175.9 | 233.0 |
| 300 | 209.2 | 6686.3 | 943.7 | 7625.9 | 147.8 | 4544.4 | 342.9 | 250.7 | 363.0 |
| 400 | 290.0 | 8937.2 | 1257.5 | 10167.0 | 192.7 | 5893.2 | 462.4 | 326.0 | 502.5 |
| 500 | 372.5 | 11189.9 | 1573.4 | 12708.9 | 245.6 | 7171.4 | 583.0 | 402.1 | 647.6 |

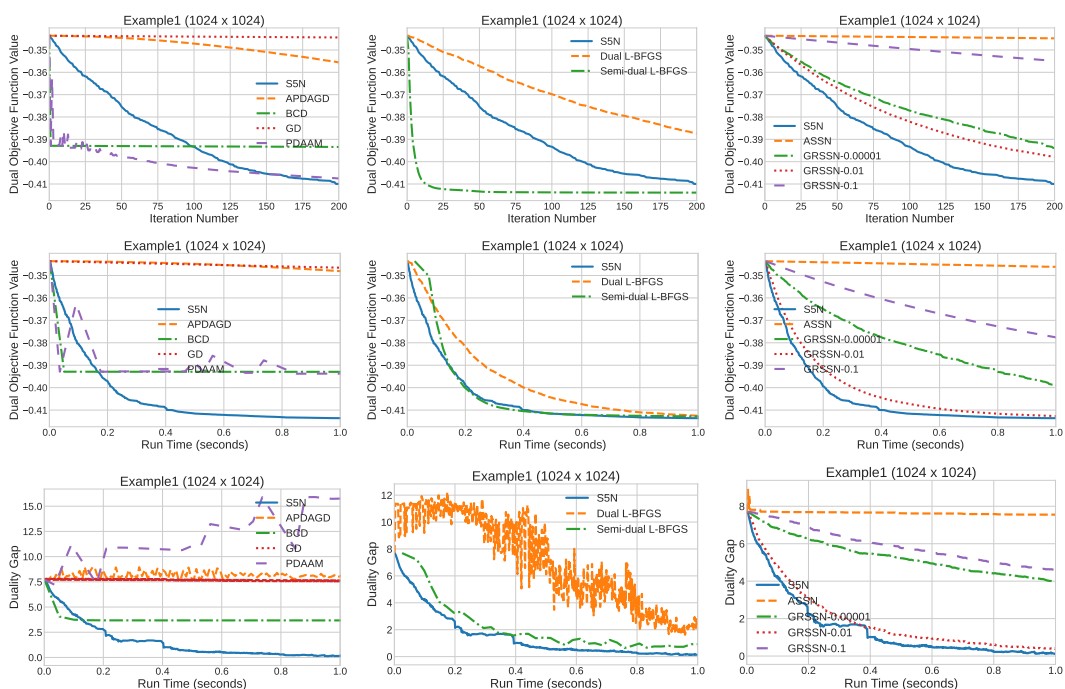

Figure 9: Example 1 ($n = m = 1024$). First row: dual objective function value vs. iteration number. Second row: dual objective function value vs. run time. Third row: duality gap vs. run time.

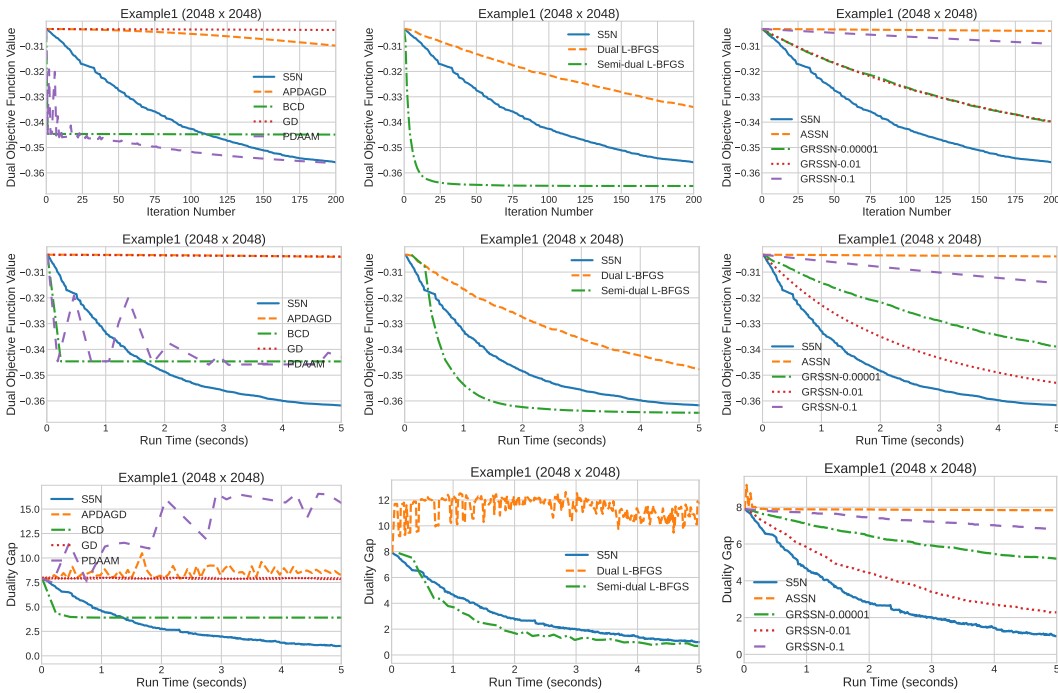

Figure 10: Example 1 ($n = m = 2048$). First row: dual objective function value vs. iteration number. Second row: dual objective function value vs. run time. Third row: duality gap vs. run time.

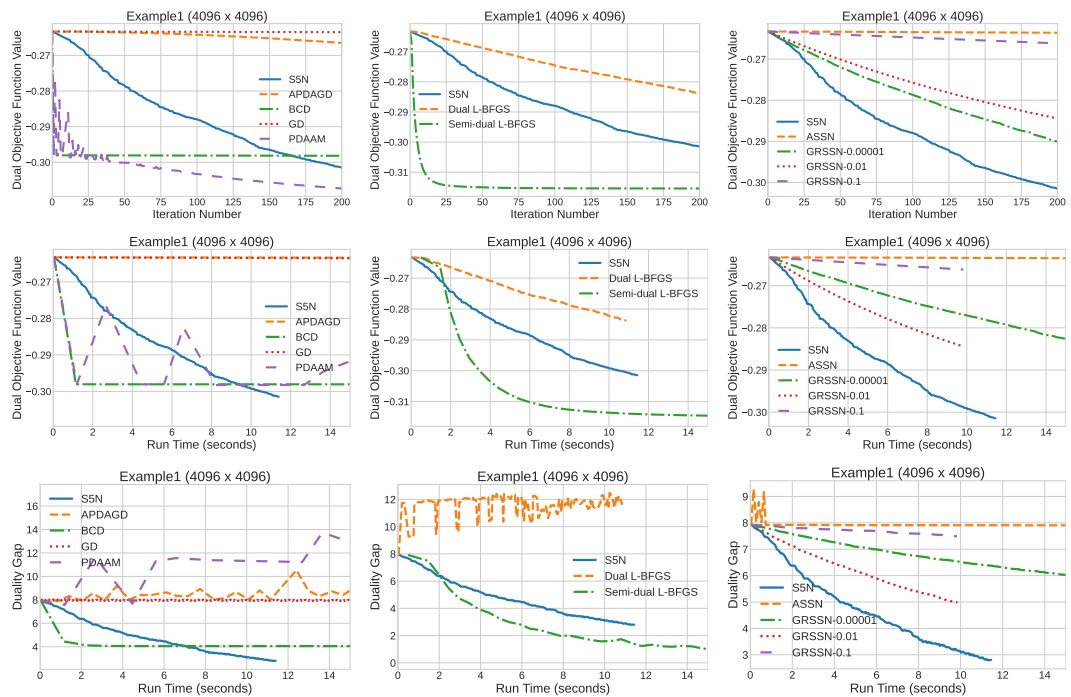

Figure 11: Example 1 ($n = m = 4096$). First row: dual objective function value vs. iteration number. Second row: dual objective function value vs. run time. Third row: duality gap vs. run time.

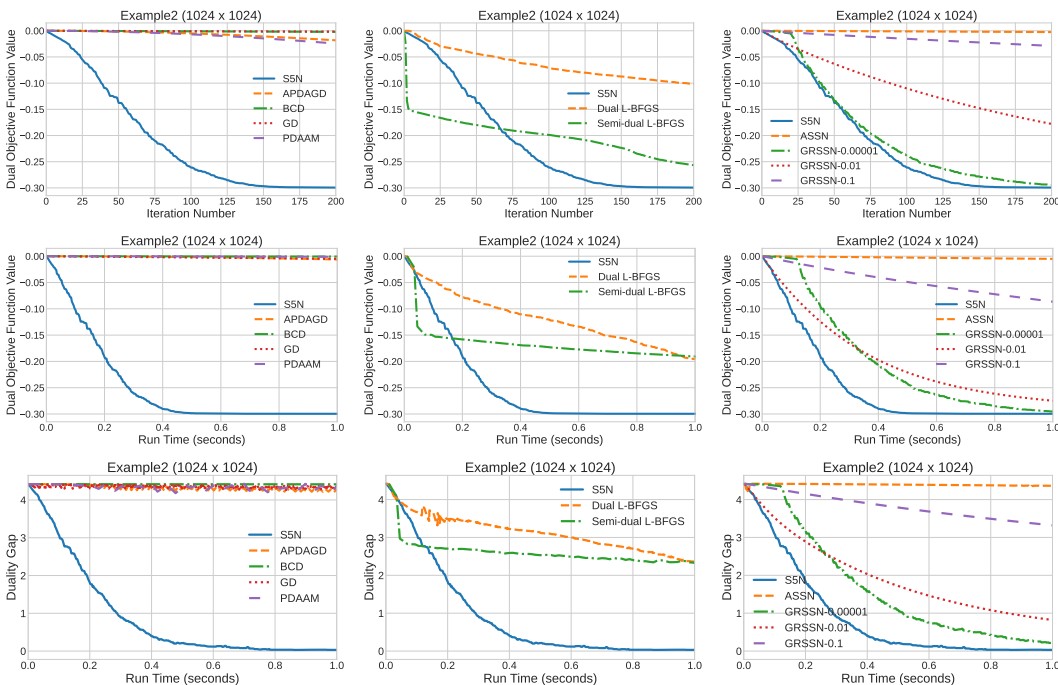

Figure 12: Example 2 ($n = m = 1024$). First row: dual objective function value vs. iteration number. Second row: dual objective function value vs. run time. Third row: duality gap vs. run time.

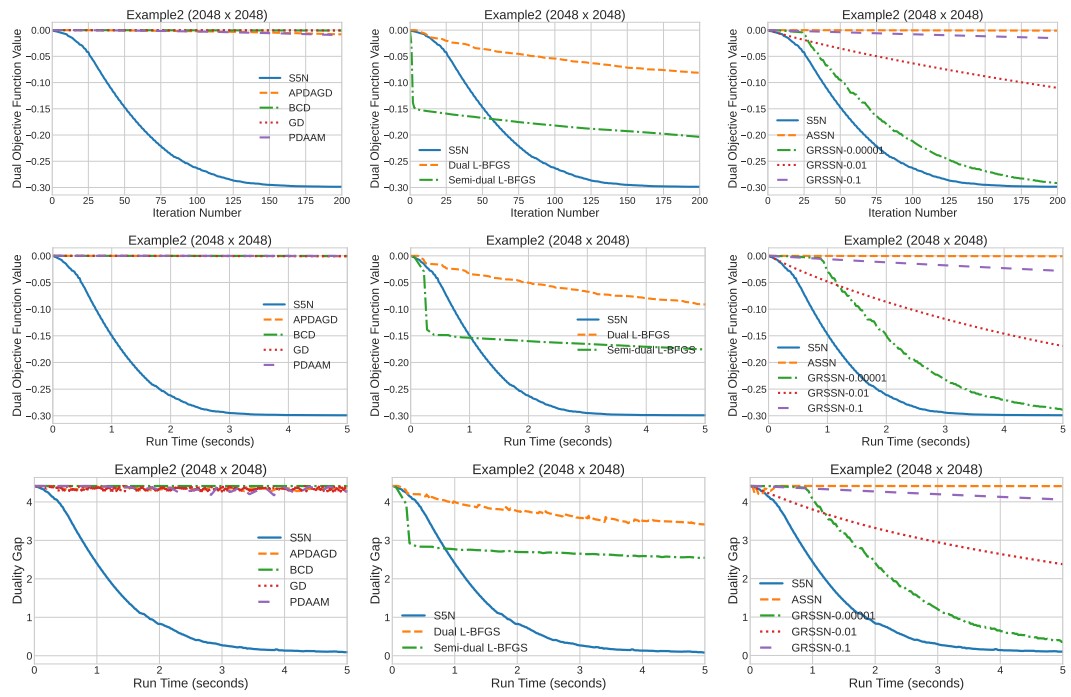

Figure 13: Example 2 ($n = m = 2048$). First row: dual objective function value vs. iteration number. Second row: dual objective function value vs. run time. Third row: duality gap vs. run time.

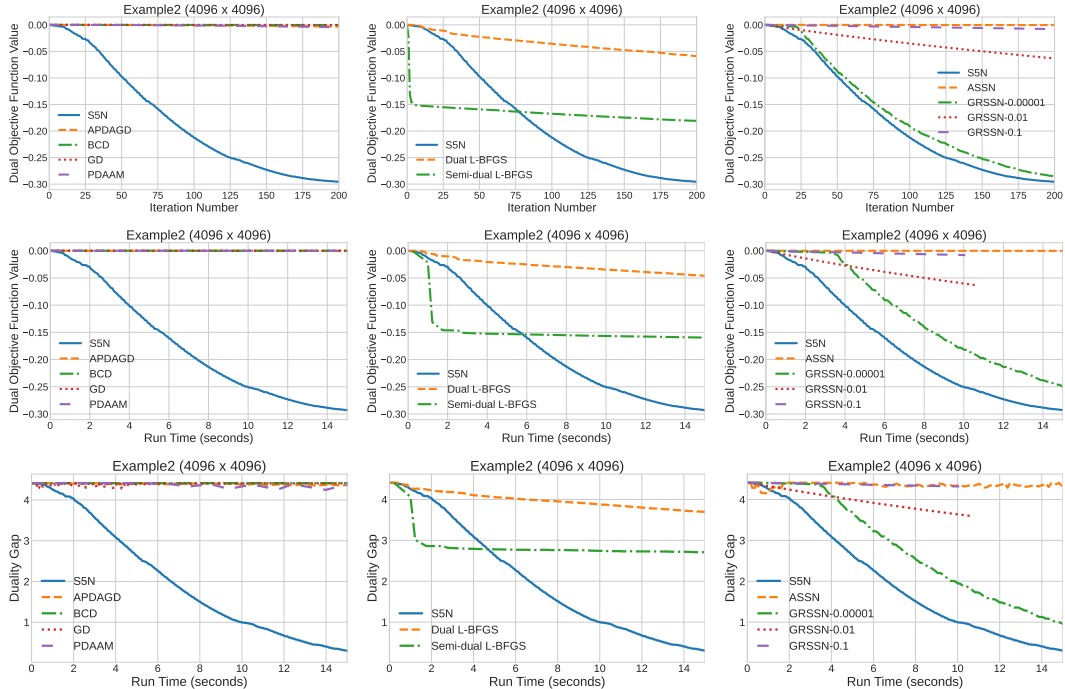

Figure 14: Example 2 ($n = m = 4096$). First row: dual objective function value vs. iteration number. Second row: dual objective function value vs. run time. Third row: duality gap vs. run time.

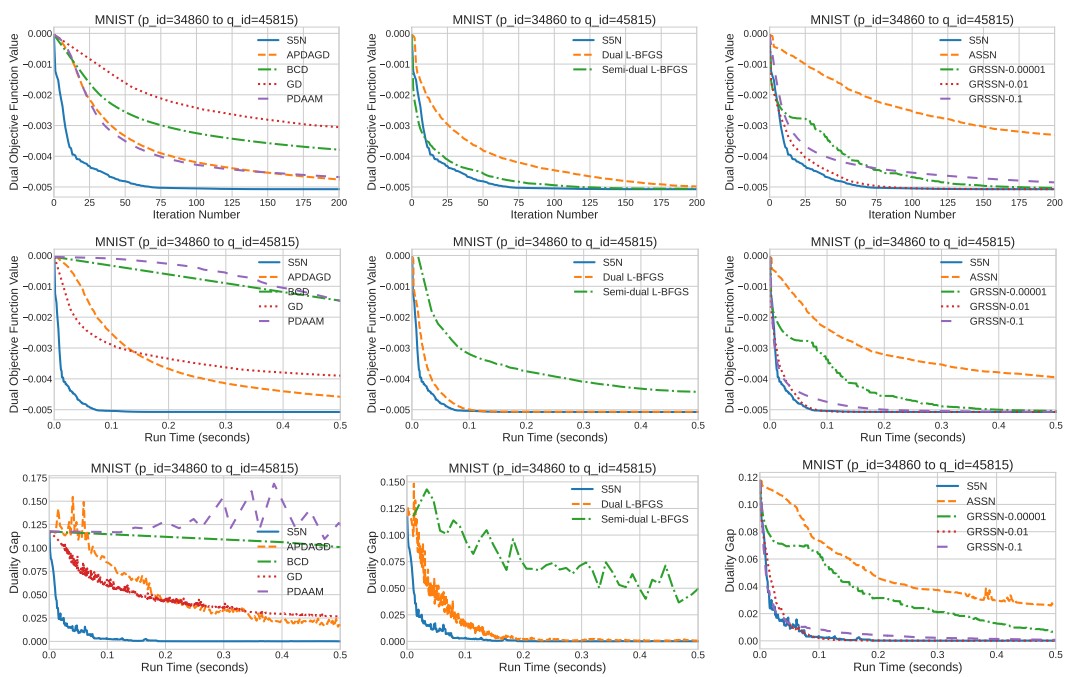

Figure 15: MNIST (image IDs 34860-45815). First row: dual objective function value vs. iteration number. Second row: dual objective function value vs. run time. Third row: duality gap vs. run time.

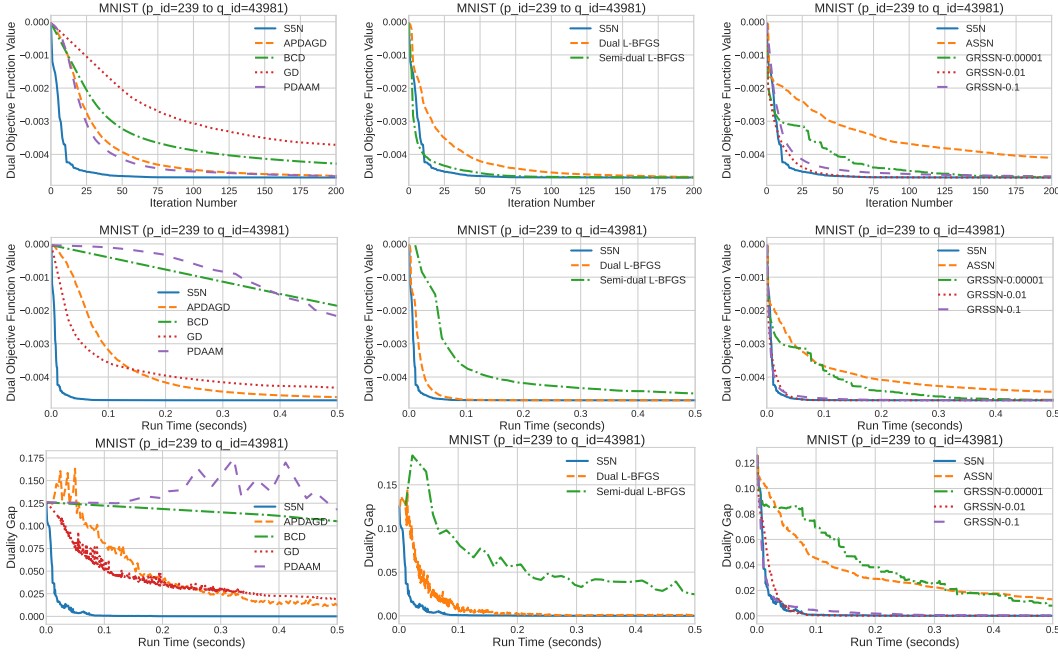

Figure 16: MNIST (image IDs 239-43981). First row: dual objective function value vs. iteration number. Second row: dual objective function value vs. run time. Third row: duality gap vs. run time.

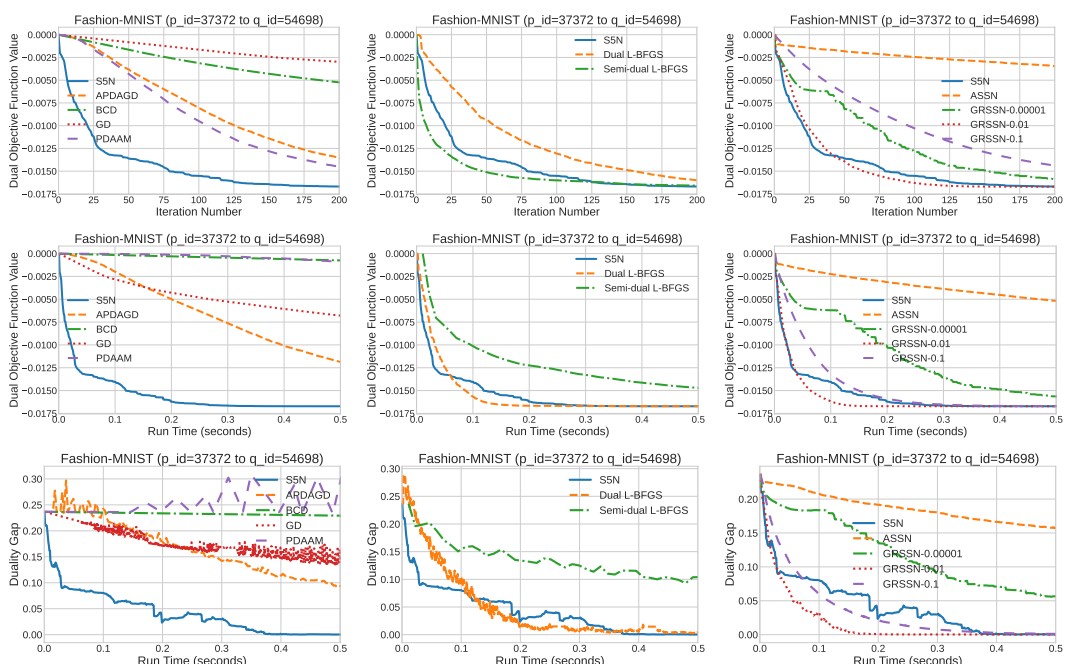

Figure 17: Fashion-MNIST (image IDs 37372-54698). First row: dual objective function value vs. iteration number. Second row: dual objective function value vs. run time. Third row: duality gap vs. run time.

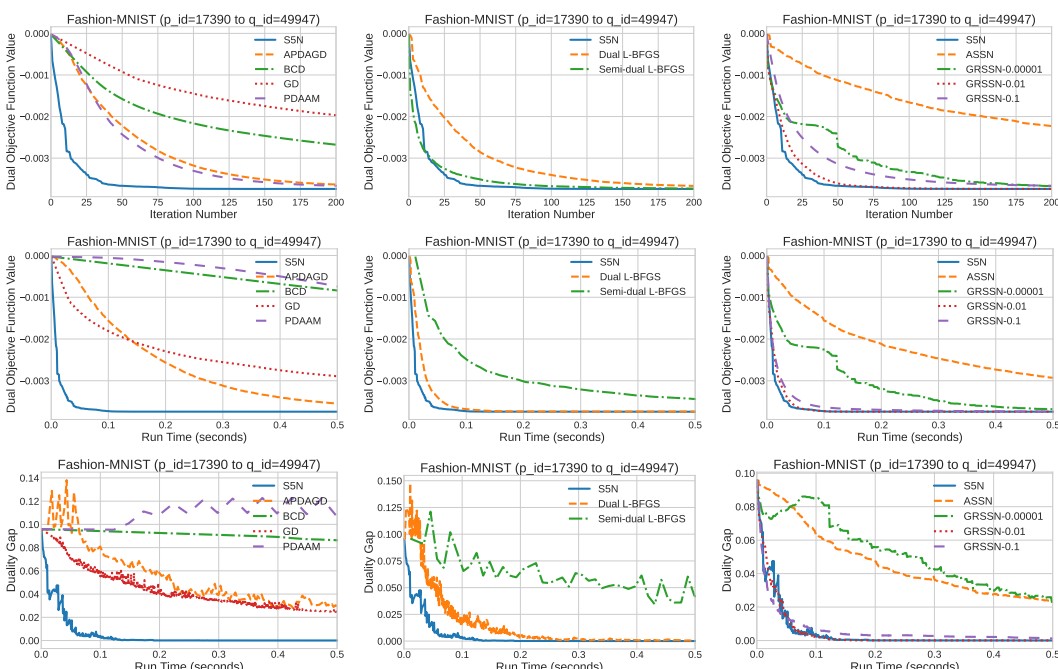

Figure 18: Fashion-MNIST (image IDs 17390-49947). First row: dual objective function value vs. iteration number. Second row: dual objective function value vs. run time. Third row: duality gap vs. run time.

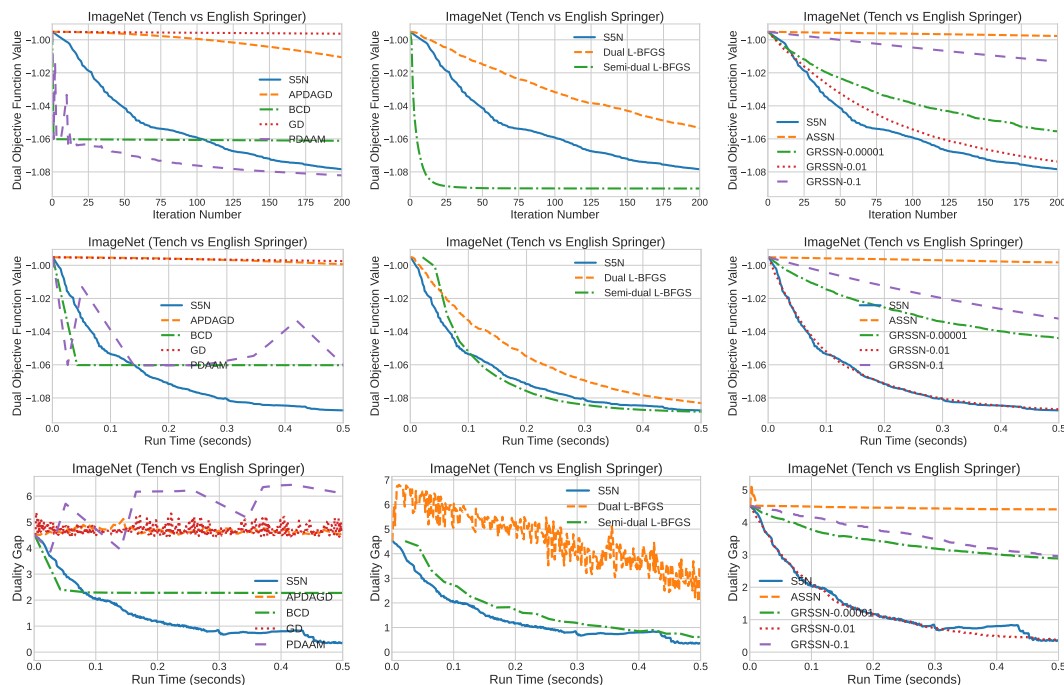

Figure 19: ImageNette (Tench-English Springer). First row: dual objective function value vs. iteration number. Second row: dual objective function value vs. run time. Third row: duality gap vs. run time.

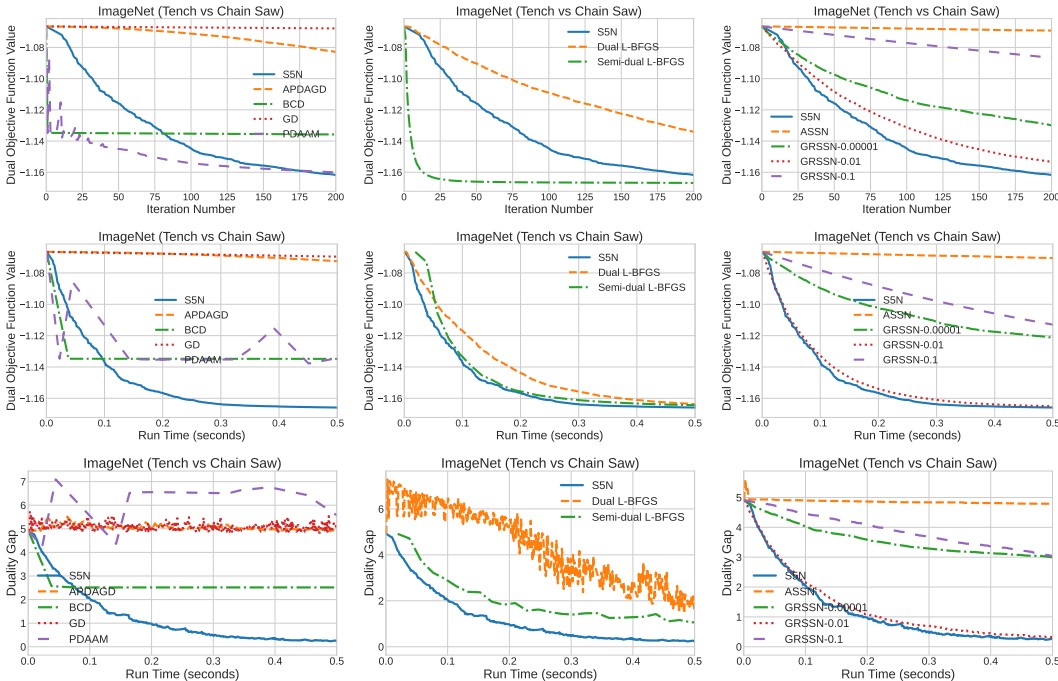

Figure 20: ImageNette (Tench-Chain Saw). First row: dual objective function value vs. iteration number. Second row: dual objective function value vs. run time. Third row: duality gap vs. run time.

