# OpenReview forum: "A Semi-smooth, Self-shifting, and Singular Newton Method for Sparse Optimal Transport"
_ICLR.cc/2024/Conference — Submitted to ICLR 2024_

### Official Review · Reviewer_35vU · 2023-10-28

**Soundness:** 3 good
**Presentation:** 3 good
**Contribution:** 3 good
**Rating:** 6
**Confidence:** 2

**Summary:**

In this paper, the authors introduce a novel Newton-type algorithm called S5N, designed to tackle problems with potentially non-differentiable gradients and non-isolated solutions. This is particularly relevant for the sparse optimal transport problem. The S5N algorithm stands out from existing Newton-type methods due to its wide applicability, absence of hyperparameter tuning, and robust global and local convergence guarantees. Numerical experiments demonstrate that S5N outperforms in terms of convergence speed and computational efficiency when applied to sparse optimal transport problems.

**Strengths:**

Some of the strengths of the paper are
* The paper is well written and the contributions are defined clearly
* In this paper the authors measure the ratio between the actual and predicted reduction of function values similar to trust-region methods to decide whether to accept the next step or not
* The proposed algorithm can handle non-smooth gradients and singular Hessian matrices
* Does not rely on line search to guarantee global convergence
* The stepsize $\eta^k$ does not need to satisfy sufficient decrease or curvature conditions as line search-based methods. It only needs to be bounded to guarantee global convergence
* The authors also show that under mild assumptions the proposed algorithm has local quadratic convergence

Thank you for providing the code!

**Weaknesses:**

I want to know why the authors used only MNIST and Fashion-MNIST apart from Examples 1 and 2 to compare their algorithm experimentally. There are plenty of different datasets that the authors could explore and share the results of how their algorithm behaves compared to other algorithms. Moreover, why did they choose only image data to test their algorithm in addition to Examples 1 and 2?

**Questions:**

Mentioned on weaknesses.

---

> ### Author Response · Authors · 2023-11-15
>
> > I want to know why the authors used only MNIST and Fashion-MNIST apart from Examples 1 and 2 to compare their algorithm experimentally. There are plenty of different datasets that the authors could explore and share the results of how their algorithm behaves compared to other algorithms. Moreover, why did they choose only image data to test their algorithm in addition to Examples 1 and 2?
>
> We appreciate the various helpful comments. Firstly, OT is not sensitive to the type of data. Its input only contains two probability vectors and a cost matrix, and all types of data are eventually summarized into these three quantities. In this sense, we are more interested in designing the input probability vectors and the cost matrix, rather than choosing the original data set. For instance, our Example 1 studies two similar distributions, and Example 2 is for two distinct distributions.
>
> The reason for choosing MNIST and Fashion-MNIST is that they have a simple and natural cost matrix, thus convenient for demonstration, enabling us to focus on algorithm development and validation. For higher-dimensional image data, it is commonly suggested that the cost matrix should be computed from some feature space rather than from the pixel space. Therefore, for more complicated image data, typically a feature extraction step is required before applying OT, which introduces another factor that needs to be controlled. Regarding this work, we focus more on the computing of sparse OT than on the application to various data types, but of course, we would be glad to explore more data sets later, once the technical questions have been properly handled.

---

> ### Author Response · Authors · 2023-11-19
> **Added experiments on ImageNet data**
>
> Dear reviewer, per the suggestion we have added more experiments on OT applied to the higher-dimensional and larger-scale ImageNet data. In particular, we compute sparse OT between two categories of images from a subset of the ImageNet data set. The subset contains ten classes of ImageNet images, and approximately 1000 images per category are selected. We map each image to a 30-dimensional feature vector by first passing the image to a ResNet18 network, resulting in a 512-dimensional vector, then followed by a dimension reduction by principal component analysis. Let $x_i\in\mathbb{R}^{30}$ be the feature vector of an image in the first category, $i=1,\ldots,n$, and $y_j\in\mathbb{R}^{30}$ be the feature vector of an image in the second category, $j=1,\ldots,m$. Then $\nu=n^{-1}\mathbf{1}_n$, $\mu=m^{-1}\mathbf{1}\_m$, and the cost matrix is  $C\_{ij}= \Vert x_i-y_j\Vert^2$.
>
> The results are displayed in Figures 19 and 20 in the revised article.

---

> > ### Comment · Reviewer_35vU · 2023-11-19
> >
> > Dear authors,
> >
> > Thank you for taking into consideration my question and adding more experiments.

---

### Official Review · Reviewer_h4ty · 2023-10-30

**Soundness:** 1 poor
**Presentation:** 2 fair
**Contribution:** 2 fair
**Rating:** 3
**Confidence:** 3

**Summary:**

The authors propose a Newton method to minimize functions with non-smooth gradients, more specifically,  differentiable functions, not differentiable twice. The main application of such an algorithm is to solve the dual of the quadratically regularized optimal transport problem. The authors provide convergence rates for the proposed algorithm and optimal transport experiments.

**Strengths:**

The algorithm seems very tailored for the considered problem.

**Weaknesses:**

- 1 In equation 2, how is the linear system solved? Do you solve it exactly (with usual decomposition techniques) or approximately, using iterative algorithms? If you use iterative algorithms, which algorithms do you use? To which tolerance / number of iterations do you solve the linear system?

- 2 The step size.
First, in the main theorem 3, no clear assumption is given on the step size $\eta_k$. I think the authors should write clearly under which assumptions the proposed algorithm converges.

Then, I do not understand how the condition on the step size (loosly given in the main text) can be so loose: "In fact, the only requirement is that $\eta_k$ needs to be
bounded", how is it possible??
If I take a sequence of exponentially decreasing step size, how can the proposed algorithm converge? This is usually why one has to resort to costly line-search techniques.
Could authors comment on this?

- 3 Experiments. I think experiments should be reshaped: I do not know how significant/useful the experiments with run time below a second are: I guess that for these timings, implementation matters much more than the algorithms themselves: I would remove experiments with run time below 1 second.
For clarity, I would select around 5 algorithms, and display all the benchmark algorithms on a single graph, and multiple values of the regularization parameter $\gamma$, like in Figure 1 of [1].
The block coordinate descent algorithm does not seem to converge, could you comment on this?

[1] Lin, H., Mairal, J. and Harchaoui, Z., 2015. A universal catalyst for first-order optimization. Advances in neural information processing systems, 28.

**Questions:**

See Weaknesses

---

> ### Author Response · Authors · 2023-11-15
> **Clarification on the step sizes**
>
> > In equation 2, how is the linear system solved? Do you solve it exactly (with usual decomposition techniques) or approximately, using iterative algorithms? If you use iterative algorithms, which algorithms do you use? To which tolerance / number of iterations do you solve the linear system?
>
> Thanks for raising this question. Equation (2) is a generic statement for solving the linear system. Specific to sparse OT, the computation method is given in Appendix C, which is based on the conjugate gradient (CG) method. Regarding the setting of precision, in our implementation, the tolerance for CG is globally set to $10^{-6}$.
>
> > The step size. First, in the main theorem 3, no clear assumption is given on the step size. I think the authors should write clearly under which assumptions the proposed algorithm converges.
>
> Our assumption on the step size is described in Section 2.2. Specifically, we require that the step sizes are bounded from both below and above: $\eta_k\in[\tilde{m},\tilde{M}]$, $0<\tilde{m}<\tilde{M}<+\infty$.
>
> > Then, I do not understand how the condition on the step size (loosely given in the main text) can be so loose: "In fact, the only requirement is that needs to be bounded", how is it possible?? If I take a sequence of exponentially decreasing step size, how can the proposed algorithm converge? This is usually why one has to resort to costly line-search techniques. Could authors comment on this?
>
> We apologize for the confusion on the term "bounded". A more precise statement is that we require the step sizes to be **bounded from both below and above**: $\eta_k\in[\tilde{m},\tilde{M}]$, $0<\tilde{m}<\tilde{M}<+\infty$. We have clarified this point in the revision to avoid confusions.
>
> > Experiments. I think experiments should be reshaped: I do not know how significant/useful the experiments with run time below a second are: I guess that for these timings, implementation matters much more than the algorithms themselves: I would remove experiments with run time below 1 second. For clarity, I would select around 5 algorithms, and display all the benchmark algorithms on a single graph, and multiple values of the regularization parameter $\gamma$ , like in Figure 1 of [1]. The block coordinate descent algorithm does not seem to converge, could you comment on this?
>
> Thanks for the suggestions. In fact, as can be inferred from Figures 1-3, for most first-order methods, they take much longer than one second to converge, and we truncate the iteration number for better illustration. Under the same setting, Newton-type methods are able to finish in shorter times.
>
> We think our timing results are comparable, as we spent lots of efforts in implementing every algorithm in efficient C++ code, and tried our best to reduce the overhead caused by programming language.
>
> For the illustration of results, since we have considered many existing algorithms, it would be visually undesirable if we put all curves in the same plot. Instead, we categorize the algorithms into three groups based on their characteristics, and then compare their performance with the S5N algorithm.
>
> For the BCD algorithm, it is normal to see a very long 'flat' stage, as this is also observed in other articles such as Figures 1 and 2 in [2]. It will converge after a sufficiently large number of iterations.
>
> [1] Lin, H., Mairal, J. and Harchaoui, Z., 2015. A universal catalyst for first-order optimization. Advances in neural information processing systems, 28.
>
> [2] Pasechnyuk, D.A., Persiianov, M., Dvurechensky, P. and Gasnikov, A., 2023. Algorithms for euclidean regularised Optimal Transport. arXiv preprint arXiv:2307.00321.

---

> > ### Comment · Reviewer_h4ty · 2023-11-21
> > **Response to Rebutal**
> >
> > I thank the authors for their time and effort in the rebuttal.
> >
> > I still think that the graphs with timing in milliseconds should be removed for graphs in larger dimension. I really believe that such timing does not reflect which algorithms is better, but much more implementation tricks
> >
> > I do not see vanilla coordinate descent in Figure 1 of [2], but a variance-reduced variation. Could authors comment on this?
> >
> > [2] Pasechnyuk, D.A., Persiianov, M., Dvurechensky, P. and Gasnikov, A., 2023. Algorithms for euclidean regularised Optimal Transport. arXiv preprint arXiv:2307.00321.

---

> > > ### Author Response · Authors · 2023-11-22
> > >
> > > Dear reviewer,
> > >
> > > Thanks for the suggestions. We have removed small-scale experiments ($n=256$) in the article, and added larger ones ($n=4096$) in Appendix D.5 (Figures 9-14). The medium-sized experiments ($n=512$) are kept, since they reflect some common practices in gradient-based learning (taking mini-batches of size around 512). They may be useful for users who want to estimate how fast each algorithm can be under a typical mini-batch computation.
> > >
> > > The block coordinate descent method in [2] is labelled as "Sinkhorn", since the Sinkhorn algorithm for entropic regularized OT is by nature a block coordinate descent algorithm.
> > >
> > > We hope the explanations above have properly addressed the previous concerns, and we are open to further discussions.

---

> ### Comment · Area_Chair_3CmY · 2023-12-05
>
> Reviewer h4ty,
>
> Thank you for the discussion with the authors. May I ask if you have any further comments? Would you keep your rating or update your evaluation?
>
> Thanks,
>
> AC

---

### Official Review · Reviewer_7Z3B · 2023-11-01

**Soundness:** 3 good
**Presentation:** 3 good
**Contribution:** 3 good
**Rating:** 6
**Confidence:** 2

**Summary:**

In this paper authors, motivated by optimal transport problems, propose Newton-type algorithm for objectives, that may have non-smooth gradients and singular Hessians. The proposed algorithm converges globally and has local quadratic convergence.

**Strengths:**

1. The paper proposes new Newton-type method, that can work with non-smooth gradients and singular Hessian matrices
2. The experimental results show, that proposed method is superior over other existing approaches
3. The results are mostly presented in a clear, understandable way.

**Weaknesses:**

1. Authors use grid-search to find the best step size on each step, which can be time-consuming in practice.
2. Proposed algorithm converges globally, but it is not clear, what is the speed of global convergence.
3. Lack of definitions of some important concepts. Maybe it is clear for those, who are very familiar with this topic, but I think, introduction of such definitions may improve the overall look of papers (see questions).

**Questions:**

1. Probably, in the abstract you have a typo and you meant "possibly non-smooth gradients" instead of "possibly non-differentiable gradients"
2. Please, increase the font size of text in the figures
3. Please, provide the definitions of local Lipschitz continuity and upper semi-continuity.

---

> ### Author Response · Authors · 2023-11-15
>
> ### Weakness part
>
> > Authors use grid-search to find the best step size on each step, which can be time-consuming in practice.
>
> Thanks for the comments. We shall point out that unlike classical line search methods, the choice of the step size in S5N is **optional**, since the theory only requires that the step sizes are bounded from both below and above. Therefore, users are allowed to set the number of candidates, and can keep it small. Also, Algorithm 2 is **not** a full grid search. It immediately stops when function value decreases, and it typically only takes one or two steps.
>
> > Proposed algorithm converges globally, but it is not clear, what is the speed of global convergence.
>
> Thanks for raising this question. To the best of our knowledge, existing global convergence results for semi-smooth Newton methods are almost all asymptotic. On the other hand, although first-order methods have non-asymptotic rates, the empirical results shown in Figures 1-3 suggest that they are much slower than Newton-type methods. We agree that the global convergence rate of semi-smooth Newton is indeed a topic that deserves future research.
>
> > Lack of definitions of some important concepts. Maybe it is clear for those, who are very familiar with this topic, but I think, introduction of such definitions may improve the overall look of papers (see questions).
>
> Thanks for the suggestions. We have added the relevant definitions in Appendix A in the revision.
>
> ### Question part
>
> > Probably, in the abstract you have a typo and you meant "possibly non-smooth gradients" instead of "possibly non-differentiable gradients"
>
> Thanks for pointing out this issue. In our context, "non-smooth" and "non-differentiable" have the same meaning. For consistency we have changed "non-differentiable" to "non-smooth" (in the PDF, as we are no longer able to edit the abstract shown in the page).
>
> > Please, increase the font size of text in the figures
>
> > Please, provide the definitions of local Lipschitz continuity and upper semi-continuity.
>
> Thanks for all these helpful suggestions. We have updated the definitions related to local Lipschitz continuity and upper semi-continuity in Appendix A, and have enlarged the font size of the text in the figures.

---

### Official Review · Reviewer_cPiV · 2023-11-04

**Soundness:** 2 fair
**Presentation:** 2 fair
**Contribution:** 3 good
**Rating:** 6
**Confidence:** 3

**Summary:**

This paper proposes a general Newton-type algorithm named
S5N, to solve problems that have possibly non-differentiable gradients and non-isolated solutions, a setting highly motivated by the sparse optimal transport problem. Compared with existing Newton-type approaches, the proposed S5N algorithm has broad applicability, does not require hyperparameter tuning, and possesses rigorous global and local convergence guarantees. Extensive numerical
experiments show that on sparse optimal transport problems, S5N gains superior performance on convergence speed and computational efficiency.

**Strengths:**

Compared with existing Newton-type approaches, the proposed S5N algorithm has broad applicability, does not require hyperparameter tuning, and possesses rigorous global and local convergence guarantees. Extensive numerical
experiments show that on sparse optimal transport problems, S5N gains superior performance on convergence speed and computational efficiency.

**Weaknesses:**

1. The results in Sec. 2.4 seems not new. They seem to be standard results of semi-smooth Newton. Thus, the author should cite some references.
2. It seems that $f(x)$ in Eq. (4) is not a smooth function because $()_+$ operator is not smooth. Thus, Proposition 1 can not hold since $\nabla f(\alpha, \beta)$ does not exist.
3. The convergence analysis of this paper lies on Assumption 1 and Assumption 2 which requires that $f(x)$ is differentiable, so the convergence analysis can not be used in the problem that $f(x)$ is not smooth.
4. This paper claims that S5N is used to solve problems that have possibly \emph{non-differentiable} gradients and non-isolated solutions.
However, the convergence analysis requires Assumption 1which supposes that $f(x)$ is differentiable.

**Questions:**

No

---

> ### Author Response · Authors · 2023-11-15
>
> First of all, we thank the reviewer for the comments, and would like to respectfully correct a factual misunderstanding in the review.
>
> > It seems that $f(x)$ in Eq. (4) is not a smooth function because $()_+$ operator is not smooth. Thus, Proposition 1 can not hold since $\nabla f(\alpha, \beta)$ does not exist.
>
> > The convergence analysis of this paper lies on Assumption 1 and Assumption 2 which requires that $f(x)$ is differentiable, so the convergence analysis can not be used in the problem that $f(x)$ is not smooth.
>
> > This paper claims that S5N is used to solve problems that have possibly *non-differentiable* gradients and non-isolated solutions. However, the convergence analysis requires Assumption 1 which supposes that $f(x)$ is differentiable.
>
> We would like to point out that $f(x)$ in Eq. (4) in fact involves $h(x)=[(x)\_+]^2$, not $(x)\_+$. It can be verified that the former **is differentiable**, with the derivative $h'(x)=2\cdot(x)_+$. Therefore, $\nabla f(\alpha,\beta)$ in Proposition 1 **indeed exists**, and Assumption 1 holds for the problem we consider.
>
> We humbly hope the reviewer could re-evaluate our main results and raise the score when appropriate, if this misunderstanding is resolved.
>
> > The results in Sec. 2.4 seems not new. They seem to be standard results of semi-smooth Newton. Thus, the author should cite some references.
>
> Thanks for the comments. While the expressions of the main conclusions in Section 2.4 appear to be standard, we have proved them for our new algorithm, S5N, under completely new settings (non-smooth gradient **and** singular Hessian). Therefore, the proof is specific to S5N, which is not a trivial consequence of existing results.
>
> We have added references to standard results, and have made our contributions clearer in the revision.

---

> > ### Comment · Reviewer_cPiV · 2023-11-16
> >
> > Since $(x)_+$ is not differentiable, why $h(x) = [(x)_+]^2$ is differentiable? Chain Rule!

---

> > > ### Author Response · Authors · 2023-11-16
> > >
> > > Dear reviewer, in some sense the chain rule is a sufficient but not necessary criterion to prove differentiability (consider $f(x)=|x|$ and $g(x)=[f(x)]^2=x^2$). For $h(x)=[(x)_+]^2$, clearly it is differentiable for $x\neq 0$, as $h(x)=0$ for $x<0$ and $h(x)=x^2$ for $x>0$. On $x=0$, the left derivative is 0, and the right derivative is $2x$, which is also zero at $x=0$. Therefore, $h'(0)=0$, and hence $h(x)$ is differentiable everywhere.

---

> > > > ### Comment · Reviewer_cPiV · 2023-11-17
> > > >
> > > > I think you are right. I raise the score to 6.

---

> > > > > ### Author Response · Authors · 2023-11-17
> > > > > **Thanks!**
> > > > >
> > > > > Thank you for the positive feedback!

---

### Official Review · Reviewer_tikC · 2023-11-09

**Soundness:** 2 fair
**Presentation:** 2 fair
**Contribution:** 2 fair
**Rating:** 3
**Confidence:** 4

**Summary:**

In the paper, the authors propose a new version of the gradient regularized Newton method with singular Hessian. It is applied to solve quadratically regularized optimal transport problems. It was proven that the method has asymptotical convergence of the gradient norm to zero. Additionally, the local quadratic convergence was proved under certain additional assumptions. The paper includes experiments with various data and setups.

**Strengths:**

The problem studied in the paper seems quite interesting, with real-life applications in optimal transport. The experimental results seem promising, and the code is implemented in an efficient, high-performance manner.

**Weaknesses:**

Unfortunately, I find it challenging to identify new and distinct results and contributions in the presented paper. Allow me to clarify my perspective. The paper consists of three main parts: 1) The S5N method for general problems with semi-smooth Hessian; 2) Application of S5N to quadratically regularized optimal transport; 3) Experiments. Next, I will discuss all of them in detail.

1) I begin with the general problems and S5N; the presented results are very similar to those in papers [1],[2]. In both papers, gradient regularization is employed with some adaptive procedures. Across all three papers, the global rates are asymptotical, and the local rates are quadratic (in 2, it is even cubic). Moreover, all these rates are relatively weak compared to modern optimization methods with non-asymptotical global convergence rates. For example, first-order methods offer guaranteed non-asymptotical global rates with a much cheaper computational cost of every iteration. Thus, from a theoretical perspective, the first part does not seem to contribute significantly compared to the existing literature from my point of view. Please correct me if I overlooked any substantial improvement and theoretical challenge over the previous papers.

2)  In the optimal transport (OT) section, the authors describe the OT problem and why it has the semi-smooth Hessian, as discussed in [3]. Therefore, the application of semi-smooth Newton to OT is also not novel (note that the authors do not claim it to be new).

3) Numerical Experiments: I appreciate the inclusion of a large number of competitors' methods.  However, the presentation and comparison seem unfair to me. I believe that comparing the Iteration Number may be biased in the context of first-order and second-order methods. I would recommend using gradient computations as a more appropriate metric for comparison. Also, theoretical per-iteration computational complexities could help. I listed more questions in the next section to clarify the methods’ performances.

To sum up, the presented contribution is not enough to be accepted for the ICLR, from my point of view.

[1] Xiantao Xiao, Yongfeng Li, Zaiwen Wen, and Liwei Zhang. A regularized semi-smooth Newton method with projection steps for composite convex programs. Journal of Scientific Computing, 76:364–389, 2018.

[2] Weijun Zhou and Xinlong Chen. On the convergence of a modified regularized Newton method for convex optimization with singular solutions. Journal of Computational and Applied Mathematics, 239:179–188, 2013.

[3] Dirk A Lorenz, Paul Manns, and Christian Meyer. Quadratically regularized optimal transport. Applied Mathematics & Optimization, 83(3):1919–1949, 2021.

**Questions:**

1) It is quite counterintuitive for me: why do we want to use second-order methods for almost quadratic problems (quadratic + linear)? Why not use simple CG or PCG to solve it?

2) Why in time plots the first-order methods are slower than S5N?

3) Why do you not compare S5N with the classical baseline for OT, the Sinkhorn method?

4) Why does the gradient time computation differ for different methods? “the semi-dual L-BFGS has a fast convergence in iteration number, it suffers from a long run time since its gradient computation is more difficult and time-consuming than other methods.”

5) Do you have any intuition: why “ASSN has a very slow convergence” in practice? It seems to have the same structure with gradient regularization.

**Details Of Ethics Concerns:**

n\a

---

> ### Author Response · Authors · 2023-11-15
> **Response to the Weakness part**
>
> > Relation to papers [1], [2]; comparison to first-order methods; improvement and theoretical challenge over the previous papers.
>
> Thanks for the comments. Regarding the two articles mentioned above, we would like to make the following clarifications:
>
> 1. [1] introduced the ASSN algorithm, which only applies to fix-point problems of the form $T(z)=z$, where $T=(1-\alpha)I+\alpha R$, and $R$ is a nonexpansive operator. This imposes a restriction on the problem to solve, whereas in contrast, S5N has substantially broader use.
> 2. In addition, the local convergence result of [1] relies on the BD-regular condition, which essentially does not support singular problems. We have reflected this point in Table 1.
> 3. Also shown in Table 1, [2] assumes that the objective function is twice continuously differentiable and the Hessian is Lipschitz continuous. These are quite strong assumptions, and that is why it can achieve local cubic convergence.
>
> In this sense, neither [1] nor [2] really applies to the general setting of non-smooth gradient and singular Hessian. In constract, we have obtained both global and local convergence properties of S5N under significantly weaker assumptions, and we think this is a major theoretical contribution that differs from existing literature.
>
> As for first-order methods, although they have non-asymptotic global convergence rates, they are mostly sub-linear, thus resulting in slower convergence speed as evidenced in Figures 1-3.
>
> More importantly, the per-iteration cost of first-order methods is not necessarily smaller than that of Newton-type methods for sparse OT problems. For example, block coordinate descent (BCD) needs to sort length-$n$ vectors $n$ times, with a total cost of $\mathcal{O}(n^2\log n)$. Newton-type methods constructs the $\sigma$ sparse matrix in Proposition 1 in $\mathcal{O}(n^2)$ time, and solves the linear equation using conjugate gradient (CG) in $\mathcal{O}(nnz\cdot\log(1/\varepsilon))$ time, where $nnz$ is the number of nonzero elements in $\sigma$, and $\varepsilon$ is the tolerance. Their total per-iteration cost can be substantially smaller than $\mathcal{O}(n^2\log n)$.
>
> > In the optimal transport (OT) section, the authors describe the OT problem and why it has the semi-smooth Hessian, as discussed in [3].
>
> Again shown in Table 1, the algorithm used by [3], named GRSSN, does not provide proofs for the global and local convergence guarantees. Crucially, GRSSN introduces a very sensitive hyperparameter $\lambda$, which we have discussed in Figures 1-3, Section 5. S5N is shown to perform better than GRSSN with no hyperparameter tuning.
>
> > Numerical Experiments
>
> To compare different algorithms, we not only consider the iteration number, but also the run time as a metric for the overall performance. We have implemented every algorithm in efficient C++ to minimize the overhead and to make the run times comparable. Also, the theoretical complexity analysis from our response above also indicates that second-order methods may even have a smaller per-iteration cost than some first-order methods in the context of sparse OT.
>
> [1] Xiantao Xiao, Yongfeng Li, Zaiwen Wen, and Liwei Zhang. A regularized semi-smooth Newton method with projection steps for composite convex programs. Journal of Scientific Computing, 76:364–389, 2018.
>
> [2] Weijun Zhou and Xinlong Chen. On the convergence of a modified regularized Newton method for convex optimization with singular solutions. Journal of Computational and Applied Mathematics, 239:179–188, 2013.
>
> [3] Dirk A Lorenz, Paul Manns, and Christian Meyer. Quadratically regularized optimal transport. Applied Mathematics & Optimization, 83(3):1919–1949, 2021.

---

> ### Author Response · Authors · 2023-11-15
> **Response to the Question part**
>
> > It is quite counterintuitive for me: why do we want to use second-order methods for almost quadratic problems (quadratic + linear)? Why not use simple CG or PCG to solve it?
>
> We would like to point out that this problem is **piecewise** quadratic, which has properties quite distinct from quadratic functions. In particular, its KKT condition is not a linear system, so CG or PCG is not applicable here.
>
> We focus on second-order methods here since (1) second-order methods in general exhibit fast convergence; (2) for sparse OT problems, the generalized Hessian is sparse, making the per-iteration cost very small.
>
> > Why in time plots the first-order methods are slower than S5N?
>
> There are two major factors: (1) Newton-type methods have faster convergence speed in terms of iteration number; (2) according to our analysis in the response above, first-order methods might even have a higher per-iteration cost. Overall, S5N exhibits visible advantage over first-order methods.
>
> > Why do you not compare S5N with the classical baseline for OT, the Sinkhorn method?
>
> Thanks for raising this question. Given that sparse OT is a highly structured problem, and Sinkhorn represents another approximation to OT that results in a dense optimal plan, our focus is on the computation of quadratically regularized OT. It is not comparable to Sinkhorn when sparsity is required.
>
> > Why does the gradient time computation differ for different methods? "the semi-dual L-BFGS has a fast convergence in iteration number, it suffers from a long run time since its gradient computation is more difficult and time-consuming than other methods."
>
> Although L-BFGS can be applied to both the dual and semi-dual objective functions, their gradient computation is different. Semi-dual L-BFGS has a similar computation as BCD, and according to our analysis above, it takes $\mathcal{O}(n^2\log n)$ time in each gradient evaluation. In contrast, dual L-BFGS is $\mathcal{O}(n^2)$. Therefore, although the total number of iterations for semi-dual might be smaller than dual, the longer per-iteration time of semi-dual results in a higher overall run time.
>
> > Do you have any intuition: why “ASSN has a very slow convergence” in practice? It seems to have the same structure with gradient regularization.
>
> We have two conjectures here: (1) although algorithmically ASSN can be applied to the QROT problem, the nature of the QROT objective function does not satisfy the convergence assumptions of the algorithm (fixed-point operator, singular Hessian, etc.); (2) from the perspective of ASSN's structure, it has no flexibility in choosing the step size. As a result, we have observed that the algorithm undergoes many rejection steps (i.e., Newton steps that do not behave well) during iterations, resulting in a slow convergence speed.

---

> > ### Comment · Reviewer_tikC · 2023-11-16
> > **Theory + Duality Gap**
> >
> > >**1:** *"[1] introduced the ASSN algorithm, which only applies to fix-point problems of the form ... This imposes a restriction on the problem to solve, whereas in contrast, S5N has substantially broader use.”*
> >
> > A) It seems to me that this statement is not true. In Sections 3.3-3.4, the authors are solving a monotone variation inequality. I haven’t seen any usage of $T(z)$ there. Monotone variation inequality is a more general problem than a convex optimization problem. Hence, It seems fair to me to compare S5N with them. Moreover, Lemma 3.7 in [1] claims that the ASSN method is monotone in terms of distance to the solution, which is stronger than just asymptotic gradient convergence.
> >
> > B) Also, for Theorem 3, it seems that $V_k$ could be any bounded positive semi-definite matrix. I haven’t found any usage of $V_k$ as a generalized Hessian. Could you provide lines in the proof of the global convergence where it is significant that $V_k$ is a generalized Hessian?
> >
> > >**2:** *“As for first-order methods, although they have non-asymptotic global convergence rates, they are mostly sub-linear, thus resulting in slower convergence speed as evidenced in Figures 1-3.”*
> >
> > I kindly disagree with this statement. From the theoretical point of view, sublinear global convergence rates are still much better and faster than global asymptotical rates. For example, from $||g_k|| \leq O(k^{-1/1000})$, one can conclude that $||g_k|| \rightarrow 0$. But the rate $O(k^{-1/1000})$ is much slower than $O(k^{-1/2})$ for the first-order methods.
> >
> > >**3:** *“More importantly, the per-iteration cost of first-order methods is not necessarily smaller than that of Newton-type methods for sparse OT problems...”*
> >
> > I was talking about the very classic first-order methods, like the Gradient Descent or Fast Gradient Method. For them, one has to calculate $g(\alpha,\beta)$ from Proposition 1. I believe for the S5N, one also has to calculate $g(\alpha,\beta)$, additionally, a sparse matrix should be constructed and CG should be performed. Hence, the iteration of S5N is not cheaper than classical first-order methods. Also, it has a worse theoretical convergence rate, which makes first-order methods dominant to S5N.
> >
> > >**4:** *“We would like to point out that this problem is piecewise quadratic, which has properties quite distinct from quadratic functions. In particular, its KKT condition is not a linear system, so CG or PCG is not applicable here.”*
> >
> > CG could be applicable for any convex and smooth function. Moreover, it has local $n$-step local quadratic convergence (The end of Section 1.3.2, Page 49 from [2]). I believe in the case of piecewise quadratic function it should move away from linear areas as the GD method and has fast convergence in quadratic areas.
> >
> > **5. Duality Gap.**
> > Do you have proof that Assumption 3 is satisfied for the problem (4)? Is the proposed method primal-dual (by solving the dual problem, the method solves the primal problem as well)? I believe for the experiments the duality gap metric is required as the main goal is to solve the primal problem, not only the dual problem.
> >
> > [1]  Xiantao Xiao, Yongfeng Li, Zaiwen Wen, and Liwei Zhang. A regularized semi-smooth Newton method with projection steps for composite convex programs. Journal of Scientific Computing, 76:364–389, 2018.
> >
> > [2] Nesterov, Yurii. Lectures on convex optimization. Vol. 137. Berlin: Springer, 2018.

---

> > > ### Author Response · Authors · 2023-11-17
> > >
> > > Dear reviewer, thanks for the detailed feedback. Below are our point-by-point responses.
> > >
> > > **1: A)** For ASSN [1], apart from requiring that $F$ is a monotone operator, the paragraph below equation (3.1) says that "we are interested in $F(z)=z-T(z)$, where $T(z)$ is a fixed-point mapping...", and Assumption 3.1 (in the final publication version, not the arXiv version) assumes that $T$ is an $\alpha$-averaged operator. Moreover, for local convergence, ASSN explicitly assumes that the generalized Hessian is non-singular (Assumption 3.5 of the publication version), so it is not completely comparable to S5N.
> > >
> > > Lemma 3.7 (Lemma 3.2 in the publication version) only applies to the projection step, and does not necessarily hold for other steps. Therefore, we think it is unfair to say that ASSN has stronger convergence result.
> > >
> > > **1: B)** This is a good finding. For brevity, we use Assumption 2 for both Theorem 3 and Theorem 4, where $V_k$ is used in the proof of local convergence. In fact, the global convergence can be proved under weaker assumptions, but we did not pursue on this direction since the generalized Hessian is eventually needed in local convergence.
> > >
> > > **2:** We also agree that a non-asymptotic global convergence rate is a nice result to pursue, but unfortunately we have not found such results in the existing literature for semi-smooth Newton, to the best of our knowledge (there are indeed non-asymptotic results for smooth and non-singular Newton). We think this to some extent indicates the difficulty of this problem. But as a guide for practical use, we are able to evaluate the convergence speed empirically, and in our experiments Newton-type methods indeed have visibly faster convergence.
> > >
> > > **3:** Note that the APDAGD algorithm we have studied in the experiments is already a type of accelerated GD method that was popular in Sinkhorn optimization. For completeness, we have just added the classical GD in all of our experiments for comparison. Of course, the classical GD has cheaper per-iteration cost than any Newton-type method, but the difference is relatively small as the linear system in Newton-type methods is very sparse. Also from empirical results, GD has slower convergence in terms of the iteration number, and the run time plots also indicate that they are slower than quasi-Newton and semi-Newton in overall performance.
> > >
> > > **4:** For nonlinear CG, the global convergence result is standard, as it relies on line search methods to select step sizes. The $n$-step local quadratic convergence is based on the condition "in a neighborhood of a **strict** minimum" (also in page 49 of [2]). However, the QROT problem in general has non-isolated solutions, so this condition does not hold for QROT. In fact, this issue inherits from the singularity of the generalized Hessian, which S5N is designed for.
> > >
> > > **5:** The key assumption is the local error bound condition (second point of Assumption 3). While a full characterization of the condition for (4) seems difficult, there are some sensible heuristics here. Since (4) is a piecewise quadratic function, it is helpful to first consider a single quadratic piece $f(x)=(1/2)x'Px-q'x$, where $P$ is positive semi-definite, but not necessarily positive definite. That is, the Hessian of $f$ can be singular. Then the solution set is the affine set $\mathcal{X}=\\{x:Px=q\\}$, and $\mathrm{dist}(x,\mathcal{X})=\Vert x - P_{\mathcal{X}}(x)\Vert=\Vert P^+(Px-q) \Vert\le \Vert P^+ \Vert\cdot\Vert Px-q \Vert$, where $P^+$ is the Moore-Penrose pseudoinverse of $P$. Note that $\Vert Px-q \Vert$ is exactly the gradient norm $\Vert g(x) \Vert$, so the local error bound condition holds as long as $P$ is not zero, even if $P$ may be singular.
> > >
> > > The paragraph under equation (4) states that an optimal solution to the primal problem can be explicitly recovered as $\Pi^* = \gamma^{-1} (\alpha^* \oplus \beta^* - C)_+$, where $(\alpha^*,\beta^*)$ is some optimal dual solution. Since all the algorithms compared in this article output dual variables, we believe they are comparable to each other.
> > >
> > > We hope the points above clarify some of the confusions, and please correct us if there is any inaccurate part.

---

> > > > ### Comment · Reviewer_tikC · 2023-11-17
> > > > **Duality Gap 2**
> > > >
> > > > **1A, 1B, 2, 4:** Thank you for the clarification and your comments.
> > > >
> > > > **3:** Thank you for the GD graphics. Finally, I found out why the time plots were so confusing to me. In Figure 1, the upper plot shows 200 iterations for every method. I assumed that the lower graphic shows the time of the same 200 total iterations for every method. That is how you can approximate time per iteration for every method. But now it seems that some methods have more iterations in the time graphic, which is confusing. So, maybe it would be better to show 200 iterations of every method for the time plots.
> > > >
> > > > **5:** When one wants to solve the primal problem $\min f(x)$ by solving the dual problem $\max g(y)$, there are some conditions to satisfy for the method. Unfortunately it is not enough to have $f(x^{\ast})=g(y^{\ast})$. The main issue is that for a convex problem, the convergence by gradient or function value does not guarantee the convergence by argument. Hence, it is possible that $||y_k-y^{\ast}||$ is still big enough, hence $||x_k-x^{\ast}||$ is big, and $f(x_k)-f^{\ast}$ is also big, which means that it does not guarantee the convergence for the primal problem. To show such convergence, one has to show either the method converges by argument to $||x_k-x^{\ast} || \leq \varepsilon$ or the method is primal-dual, which means the duality gap $f(x_k)-g(y_k)\leq \varepsilon$. In the experimental setup, it means that one either shows the graphic for $||y_k-y^{\ast}||$, which is hard to do, or $f(x_k)-g(y_k)$, which is much easier and it shows that the method solves both primal and dual problems.

---

> > > > > ### Author Response · Authors · 2023-11-19
> > > > > **Added summarization of computing time and duality gap plots**
> > > > >
> > > > > Dear reviewer, thanks for the helpful comments.
> > > > >
> > > > > Regarding the run time plots, that was actually our original scheme, but later we found that the curve for GD was too short due to its low per-iteration cost. So this is our current arrangement: we still run each algorithm long enough to make the run time plot, but show the time vs. iteration in Tables 2-4 newly added in the revision. In this way the per-iteration cost can be better estimated.
> > > > >
> > > > > We also thank for the explanation on duality gap. We have updated the code to compute duality gaps for all our experiments, and the plots for the examples in the main text are given in Figures 6-8.

---

> > > > > ### Author Response · Authors · 2023-11-22
> > > > >
> > > > > Dear reviewer,
> > > > >
> > > > > We would like to check if there are any further concerns about our article and rebuttal. We are glad to answer additional questions if there is anything unclear. Also, we humbly hope you could consider updating the overall rating if the previous issues have been fixed.
> > > > >
> > > > > Thank you for your support.

---

> ### Comment · Area_Chair_3CmY · 2023-12-05
>
> Reviewer tikC,
>
> Thank you for the discussion with the authors. May I ask if you have any further comments? Would you keep your rating or update your evaluation?
>
> Thanks,
>
> AC

---

### Author Response · Authors · 2023-11-23
**Summary of updates in the revision**

Dear Area Chairs and Reviewers,

We are immensely thankful for your constructive feedback and the engaging discussions during the review process, which greatly improved the quality of our article.

Regarding our method S5N, it is a Newton-type optimization method specifically designed to handle first-order continuously differentiable but second-order non-differentiable objective functions, and it also allows for singular Hessian matrices that lead to non-isolated solutions. As an important application, S5N is effective in solving sparse optimal transport (OT) problems that are commonly seen in machine learning tasks.

Following your valuable suggestions, we have made several enhancements to our paper:

1. Through the various discussions, we have made it clear that the proposed S5N is a novel method capable of solving a broad class of optimization problems that existing methods are not applicable (non-smooth gradient, singular Hessian, etc.), both in the algorithmic sense and in the theoretical sense.

2. In response to Reviewer tikC's suggestion, who highlighted the importance of demonstrating the duality gap as a key metric for primal-dual problems, we have updated our code to compute the duality gap during the optimization process, and have added illustrations showing the duality gap's relationship with time and iteration for our experiments. From the perspective of duality gaps, our method shows clear advantages over existing algorithms. Additionally, to avoid potential misinterpretation from the initial time comparison plots, we have redrawn these figures and included tables (Tables 2-4) detailing the relationship between the number of iterations and the run time for different algorithms, aiming for a clearer demonstration of our method's performance.

3. In response to Reviewer h4ty's comments, we have replaced the 256-dimensional experiments with 4096-dimensional ones and increased the iteration limits (Figures 11, 14), which further reduces the impact of software implementation. The final results demonstrate S5N's significant advantage in large datasets.

4. Addressing Reviewer 35vU's concern about the persuasiveness of experiments using only simulated data and MNIST-like data, we have added more experiments on OT applied to the higher-dimensional and larger-scale ImageNet data (Figures 19-20). Likewise, S5N shows visible advantages on more complex real data.

5. Thanks to the comments by Reviewer cPiV and Reviewer 7Z3B, we have made clarifications on some important definitions in the theory and key parameters of the algorithm, which we hope minimize the confusions in the previous manuscript.

Beyond these improvements to the paper, your insightful comments and suggestions have been duly noted. We will thoughtfully consider these points and strive to address these areas in our future research.

As we approach the end of the discussion phase, we would like to extend an invitation to Reviewers tikC and h4ty to verify if their initial reservations have been satisfactorily addressed through our revisions. It is our hope that the amendments align with your expectations. Please feel free to raise any additional queries or concerns you might have, as we stand ready to engage and refine our work further to enhance the quality of our submission.

We are immensely grateful to the reviewers' dedication and the time invested in scrutinizing our work, which has been instrumental in its refinement.

---

### Meta-Review · Area_Chair_3CmY · 2023-12-06

**Metareview:**

In their paper, the authors present a new Newton-type algorithm and consider its application on sparse optimal transportation problem. This method distinguishes itself from other Newton-type algorithms through its broad applicability, lack of need for hyperparameter adjustments, and strong assurances of both global and local convergence.

The review team have raise several concerns on this paper. From my perspective, the important issues include:

1. The global convergence rate of the proposed algorithm is unknow. In particular, it is unclear when the algorithm will reach the local convergence area and start to have the proved local convergence rate.

2. The local convergence is proved under unverifiable assumptions for OT. So, these assumptions are possibly not satisfied for the OT problem.

3. The algorithm needs to solve a linear system in each iteration which can be time consuming when the dimension is high.

4. In the implementation, the linear system is only solved inexactly so there is a small error. However, this error is not considered in the convergence analysis.

**Justification For Why Not Higher Score:**

Please see the Metareview

**Justification For Why Not Lower Score:**

N/A

---

### Decision · Program_Chairs · 2024-01-16

Reject